# The paraventricular thalamus is a critical mediator of top-down control of cue-motivated behavior in rats

**Paolo Campus[1][†], Ignacio R Covelo[1][†][‡], Youngsoo Kim[2], Aram Parsegian[1], Brittany N Kuhn[3], Sofia A Lopez[3], John F Neumaier[4], Susan M Ferguson[4], Leah C Solberg Woods[5], Martin Sarter[2,3], Shelly B Flagel[1,6][§]***

[1]Molecular and Behavioral Neuroscience Institute, University of Michigan, Ann Arbor, United States; [2]Department of Psychology, University of Michigan, Ann Arbor, United States; [3]Neuroscience Graduate Program, University of Michigan, Ann Arbor, United States; [4]Department of Psychiatry and Behavioral Sciences, University of Washington, Seattle, United States; [5]Department of Internal Medicine, Section on Molecular Medicine, Wake Forest School of Medicine, Winston-Salem, United States; [6]Department of Psychiatry, University of Michigan, Ann Arbor, United States

**\*For correspondence:**
sflagel@umich.edu

[†]These authors contributed equally to this work

**Present address:** [‡]Department of Psychology, University of Wisconsin-Parkside, Kenosha, United States; [§]Molecular and Behavioral Neuroscience Institute, University of Michigan, Ann Arbor, United States

**Competing interests:** The authors declare that no competing interests exist.

**Abstract** Cues in the environment can elicit complex emotional states, and thereby maladaptive behavior, as a function of their ascribed value. Here we capture individual variation in the propensity to attribute motivational value to reward-cues using the sign-tracker/goal-tracker animal model. Goal-trackers attribute predictive value to reward-cues, and sign-trackers attribute both predictive and incentive value. Using chemogenetics and microdialysis, we show that, in sign-trackers, stimulation of the neuronal pathway from the prelimbic cortex (PrL) to the paraventricular nucleus of the thalamus (PVT) decreases the incentive value of a reward-cue. In contrast, in goal-trackers, inhibition of the PrL-PVT pathway increases both the incentive value and dopamine levels in the nucleus accumbens shell. The PrL-PVT pathway, therefore, exerts top-down control over the dopamine-dependent process of incentive salience attribution. These results highlight PrL-PVT pathway as a potential target for treating psychopathologies associated with the attribution of excessive incentive value to reward-cues, including addiction.
DOI: https://doi.org/10.7554/eLife.49041.001

## Introduction

Learning to associate environmental stimuli with the availability of valuable resources, such as food, is critical for survival. Such stimulus-reward associations rely on Pavlovian conditioning, a learning process during which a once neutral stimulus becomes a conditioned stimulus (CS), as it reliably predicts the delivery of an unconditioned stimulus (US) (e.g. food). The CS, then, attains predictive value and comes to elicit a conditioned response. Yet, we know from both preclinical and clinical studies that CSs can also acquire incentive value and elicit complex emotional and motivational states (*Robinson and Berridge, 2008*; *Robinson and Flagel, 2009*; *Tibboel et al., 2015*; *Pool et al., 2016*). When a CS is attributed with incentive salience and transformed into an incentive stimulus, it becomes attractive and desirable in its own right. That is, the CS becomes a 'motivational magnet', now capable of capturing attention and eliciting approach behavior (*Berridge, 2009a*). However, individuals vary in their propensity to attribute incentive value to reward cues, and only for some individuals do such cues acquire inordinate control over behavior and the ability to elicit maladaptive tendencies (*Flagel et al., 2007*; *Flagel et al., 2009*) that are characteristic of psychopathology.

Indeed, several psychiatric disorders have been associated with the excessive attribution of motivational significance to environmental cues, including substance use disorder (*Robinson and Berridge, 1993*; *Berridge and Robinson, 2016*; *Kwako et al., 2017*; *MacNiven et al., 2018*), eating disorders (*Berridge et al., 2009b*; *Robinson, 2014*), gambling disorder (*Limbrick-Oldfield et al., 2017*), post-traumatic stress disorder (PTSD) (*Coffey et al., 2010*), and bipolar disorder (*Mason et al., 2012*; *Whitton et al., 2015*).

In recent years, an animal model has been established that enables us to parse the neurobiological mechanisms that may bias the way an individual responds to reward-cues. While the preferential use of predictive vs. incentive learning strategies may be adaptive under the right conditions; an extreme bias for the selective use of a single strategy may contribute to increased risk for psychopathology. When rats are exposed to a Pavlovian conditioned approach (PavCA) paradigm in which the presentation of a lever-CS is immediately followed by the response-independent delivery of food-US, distinct conditioned responses emerge. Some rats, goal-trackers (GTs), approach the location of impending food delivery upon the lever-CS presentation, while others, sign-trackers (STs), approach and interact with the lever-CS itself. For both GTs and STs the lever-CS acquires predictive value and elicits a conditioned response, but for STs, the CS also acquires incentive value. This animal model, therefore, provides a unique platform to investigate the neurobiological determinants of individual differences in the propensity to attribute incentive salience to reward-cues.

Previous studies suggest that sign-tracking behavior results from enhanced activity in subcortical brain circuits known to mediate motivated behaviors, including the striatal dopamine system, the amygdala, and the hypothalamus (*Flagel et al., 2011a*; *Flagel et al., 2011b*; *Saunders and Robinson, 2012*; *Yager et al., 2015*; *Singer et al., 2016*; *Haight et al., 2017*). In addition, relative to GTs, STs appear to have deficits in top-down cognitive control originating in the prefrontal cortex (*Paolone et al., 2013*). Thus, we hypothesize that sign-tracking behavior arises from an imbalance between top-down cognitive control and bottom-up motivational processes. One brain region that is ideally situated to act as a fulcrum between cortical, limbic and homeostatic circuits is the paraventricular nucleus of the thalamus (PVT). The PVT receives cortical afferents from the medial PFC, including the infralimbic (IL) and prelimbic (PrL) cortices, and subcortical afferents from the hypothalamus, amygdala, and several brainstem regions involved in visceral functions and homeostatic regulation (*Hsu and Price, 2007*; *Li and Kirouac, 2012*; *Kirouac, 2015*). The PVT sends projections to various brain regions that have been associated with reward-learning and motivated behaviors, including the PrL and IL cortices, nucleus accumbens (NAc) core (NAcC) and shell (NAcS), lateral bed nucleus of the stria terminalis and central amygdala (*Hsu and Price, 2007*; *Li and Kirouac, 2008*; *Hsu and Price, 2009*). Recent findings surrounding the functional role of these PVT circuits (*Do-Monte et al., 2015*; *Haight et al., 2017*; *Millan et al., 2017*; *Giannotti et al., 2018*) have garnered recognition of the PVT as the 'thalamic gateway' (*Millan et al., 2017*) for appetitive motivation; acting to integrate cognitive, emotional, motivational and viscerosensitive information, and, in turn, guide behavioral responses (*Kirouac, 2015*; *Millan et al., 2017*). Consistent with this view, the PVT has been implicated in the propensity to attribute incentive motivational value to reward-cues (*Haight and Flagel, 2014*; *Haight et al., 2015*; *Kuhn et al., 2018*).

The functional connectivity of the PVT in response to cue-induced neuronal activity differentiates STs from GTs (*Flagel et al., 2011a*; *Haight and Flagel, 2014*; *Haight et al., 2017*). In STs, cue-induced activity in the PVT is correlated with activity in subcortical areas, including the NAc; whereas in GTs, cue-induced activity in the PVT is correlated with activity in the PrL (*Flagel et al., 2011a*; *Haight and Flagel, 2014*). Further investigation of PVT-associated circuitry in STs and GTs revealed that these phenotypes exhibit the same degree of cue-induced neural activity in PrL neurons that project to the PVT; but STs show greater cue-induced activity in subcortical afferents to the PVT, including the hypothalamus, and efferents from the PVT, including those to the NAc (*Haight et al., 2017*). These data suggest that the predictive value of the reward-cue is encoded in the PrL-PVT circuit. In STs, however, cognitive information about the predictive value of the reward-cue presumably competes with overriding subcortical motivational circuits, thus rendering them more prone to attribute incentive value. From this, we hypothesized that stimulating the PrL-PVT circuit (i.e. enhancing top-down control) in STs would reduce the tendency to attribute incentive value to a food-cue, by counteracting their inherent bias towards bottom-up motivational mechanisms. In contrast, we hypothesized that inhibiting the PrL-PVT circuit (i.e. attenuating top-down control) in GTs would increase the tendency to attribute incentive value to a food-cue, by weakening the top-down

cognitive component of the system and permitting bottom-up motivational mechanisms to act. Because the PVT sends dense projections to the NAc (*Berendse and Groenewegen, 1990*; *Li and Kirouac, 2008*; *Kirouac, 2015*), and can affect local dopamine (DA) release (*Jones et al., 1989*; *Pinto et al., 2003*; *Parsons et al., 2007*; *Choi et al., 2012*; *Perez and Lodge, 2018*), which is critical for incentive learning (*Berridge and Robinson, 1998*; *Flagel et al., 2011b*; *Saunders and Robinson, 2012*), we also hypothesized that manipulations of PrL-PVT activity would affect extracellular DA levels in the NAcS, where PVT connections are most dense (*Li and Kirouac, 2008*). Specifically, we predicted that stimulation of the PrL-PVT circuit in STs would decrease DA, whereas inhibition of the PrL-PVT circuit in GTs would increase DA in the NAcS. To test these hypotheses, we used a dual-vector approach (*Soudais et al., 2001*; *Boender et al., 2014*; *Kerstetter et al., 2016*) to express either the stimulatory Gq- or inhibitory Gi/o- DREADD (Designer Receptors Exclusively Activated by Designer Drugs) in neurons of the PrL that project to the PVT, and examined how bidirectional manipulations of this pathway affect the attribution of incentive salience to a food-cue (Experiment 1; *Figure 1*). In addition, we used in-vivo microdialysis to assess extracellular levels of DA in the NAcS following manipulations of the PrL-PVT pathway (Experiment 2, Figure 5a–f).

## Results

### Experiment 1

#### Acquisition of pavlovian conditioned approach behaviors

The average PavCA index from sessions 4–5 was used to classify rats as STs (PavCA index $\geq$ +0.30) or GTs (PavCA index $\leq -0.30$) (*Figure 2b*). As explained in the Methods below, the intermediate population of rats (PavCA index between $-0.30$ and +0.30) were excluded from this study. Linear mixed-effects models revealed a significant effect of phenotype, session and a significant phenotype x session interaction for all measures of sign- and goal-tracking behavior. Across the 5 sessions of PavCA training, STs had a greater number of lever contacts ($F_{4,184.225}$=57.778, p<0.001), a greater probability to contact the lever ($F_{4,303.698}$=68.278, p<0.001), and a lower latency to approach the lever ($F_{4,336.578}$=63.676, p<0.001) (*Figure 2c–e*). These significant differences were apparent during sessions 1–5 of PavCA training (post-hoc analyses, p<0.001). In contrast, across the 5 sessions of PavCA training, GTs showed a greater number of magazine entries ($F_{4,243.952}$=57.436, p<0.001), a greater probability to enter the magazine ($F_{4,225.359}$=76.968, p<0.001), and a lower latency to enter the magazine ($F_{4,348.976}$=71.788, p<0.001) (*Figure 2f–h*), and these significant differences were apparent during PavCA sessions 2–5 (post-hoc analyses, p<0.05).

Experimental groups (i.e. G-protein coupled receptor (GPCR) and treatment groups) were counterbalanced based on the average PavCA index from sessions 4–5 (*Figure 2i*). A three-way ANOVA (phenotype x GPCR x treatment) showed a significant effect of phenotype ($F_{1,114}$=2685.054, p<0.001) on PavCA index (*Figure 2i*), but no significant effects of GPCR or treatment groups, and no significant interactions (see also *Supplementary file 1* and *2*).

#### PavCA rescreening vs. PavCA test

PavCA index is presented as the primary dependent variable in the main text, but analyses for other dependent variables indicative of Pavlovian conditioned approach behavior including, contacts, probability and latency directed towards either the lever-CS or food magazine are included in *Supplementary file 3* and *4*. PavCA index during each daily session of rescreening is presented in *Figure 3—figure supplement 1*.

#### Stimulation of the PrL-PVT pathway attenuates the incentive value of the food cue in STs

Stimulation of the PrL-PVT pathway in STs significantly decreased the PavCA index (*Figure 3c*), which, in this case, is reflective of both a decrease in lever-directed behaviors (*Supplementary file 3*) and an increase in goal-directed behaviors (*Supplementary file 4*). There was a significant effect of treatment ($F_{1,23}$ = 33.251, p<0.001), session ($F_{1,23}$ = 10.799, p = 0.03), and a significant treatment x session interaction ($F_{1,23}$ = 14.051, p = 0.01, 1-β = 1) for the PavCA index. Post-hoc analyses revealed a significant difference between VEH- and CNO-treated STs during both rescreening

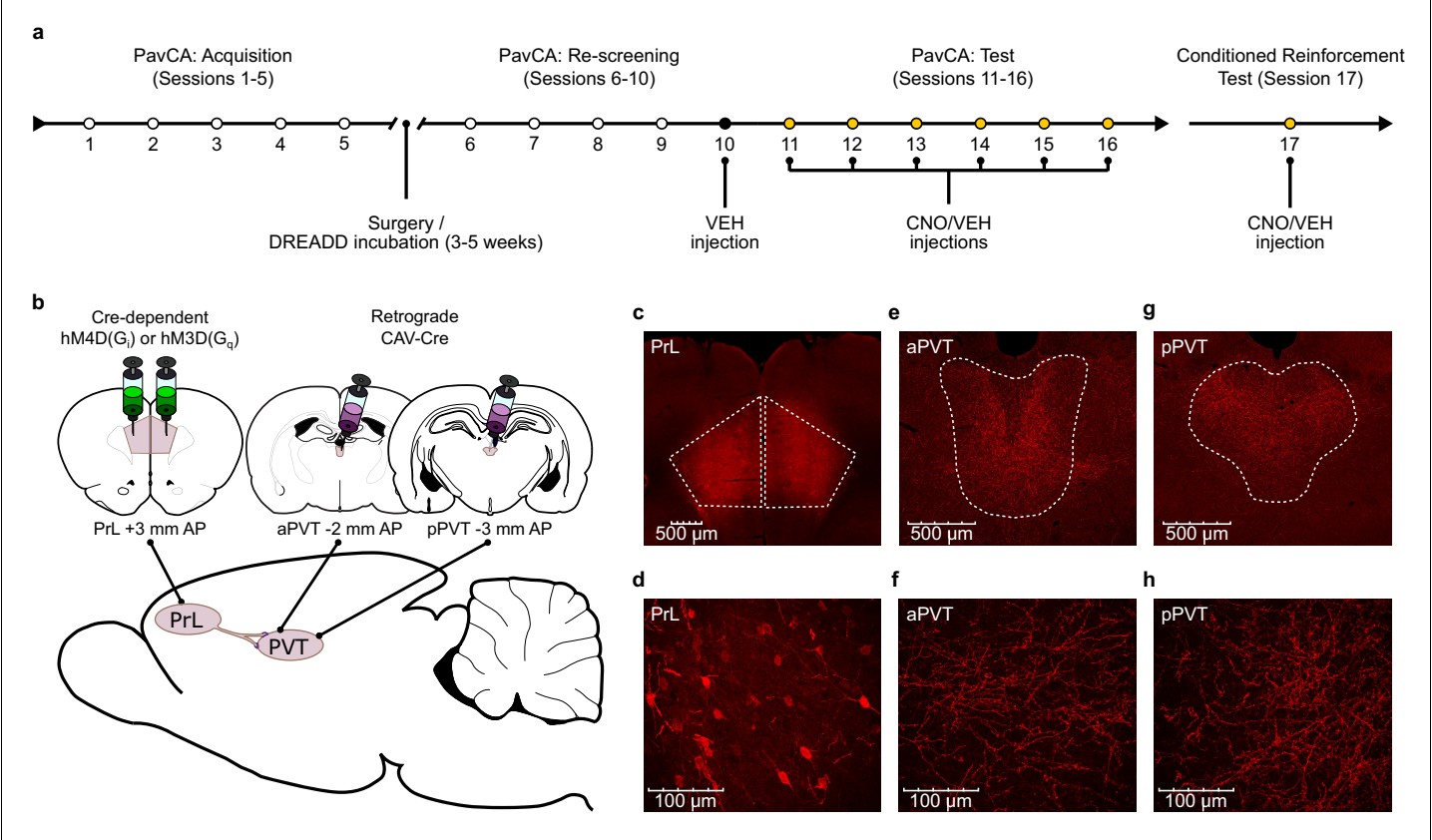

**Figure 1.** Experiment 1 methods. (**a**) Timeline of the experimental procedures. Rats were trained in a Pavlovian Conditioned Approach (PavCA) paradigm for five consecutive days (Acquisition, Sessions 1–5) and phenotyped as sign- (STs) or goal-trackers (GTs). Following acquisition, STs and GTs underwent DREADD surgeries for delivering Gi- or Gq- DREADDs in neurons of the prelimbic cortex (PrL) projecting to the paraventricular nucleus of the thalamus (PVT). Incubation time for DREADD expression was 3–5 weeks. After incubation, rats were re-screened for sign- and goal-tracking behavior (Re-screening, Sessions 6–10). All rats received an i.p. VEH injection 25 min before session 10 to habituate them to the injection procedure. CNO (3 mg/kg) or VEH were then administered i.p. every day during the Test phase (Sessions 11–16), 25 min before the start of each session. 24 hr after the last session of PavCA, rats received an additional injection of CNO or VEH 25 min before being exposed to a Conditioned Teinforcement Test (CRT, Session 17). (**b**) Schematic of the dual-vector strategy used for expressing Gi- or Gq- DREADDs in the PrL-PVT pathway. (**c,d**) Photomicrographs representing mCherry expression in pyramidal neurons of the PrL projecting to the PVT at (**c**) 4x magnification and (**d**) 40x magnification. (**e,f**) Photomicrographs of mCherry expression representing terminal fibers in the anterior PVT coming from the PrL at (**e**) 10x magnification and at (**f**) 40x magnification. (**g,h**) Photomicrographs of mCherry expression representing terminal fibers in the posterior PVT coming from the PrL at (**g**) 10x magnification and (**h**) 40x magnification.

DOI: https://doi.org/10.7554/eLife.49041.002

The following figure supplement is available for figure 1:

**Figure supplement 1.** Representative photomicrographs of mCherry expression.

DOI: https://doi.org/10.7554/eLife.49041.003

(p=0.005, Cohen's d = 1.27) and test (p<0.001, Cohen's d = 2.74). The significant difference between treatment groups during rescreening, prior to actual treatment, is due, in part, to the fact that counterbalancing was disrupted once animals were eliminated because of inaccurate DREADD expression. Importantly, however, only the CNO-treated rats exhibited a change in behavior during the test sessions relative to rescreening (p<0.001, Cohen's d = 1.18). For GTs (*Figure 3d*), there was not a significant effect of treatment ($F_{1,10}$ = 0.169, p = 0.690), session ($F_{1,10}$ = 0.511, p = 0.491) nor a significant treatment x session interaction ($F_{1,10}$ = 0.351, p = 0.567). The modest sample size (n = 6) may have contributed to the lack of effects in the GT-Gq rats, as suggested by a post-hoc power analysis (1-β = 0.27). Nonetheless, taken together, these results suggest that 'turning on' the top-down PrL-PVT circuit appears to selectively attenuate the incentive value of the cue in STs.

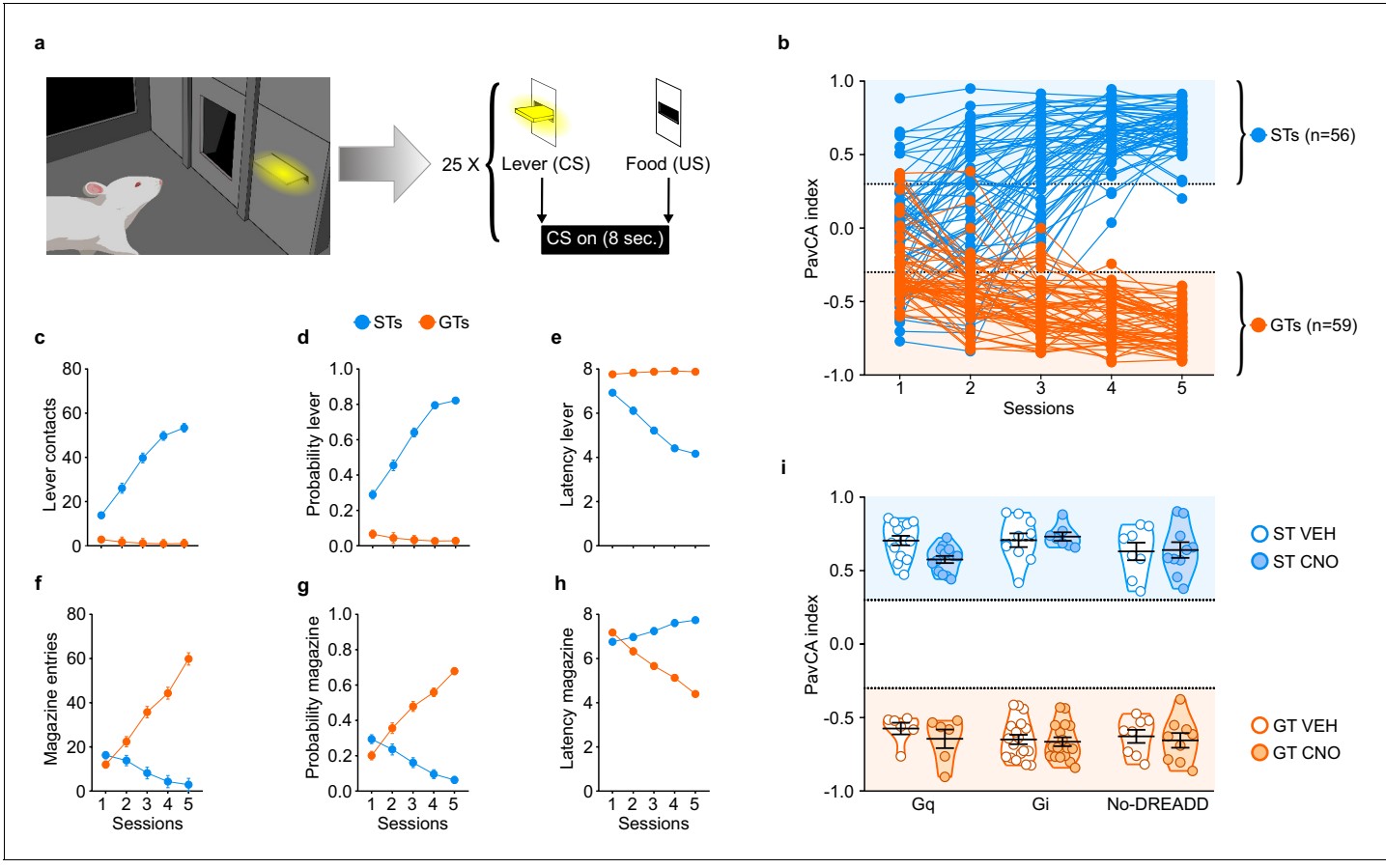

**Figure 2.** Acquisition of sign- and goal-tracking behaviors during 5 sessions of Pavlovian Conditioned Approach (PavCA) training (i.e. prior to surgery or CNO administration). (**a**) Schematic representing the PavCA task. Rats were presented with an illuminated lever (conditioned stimulus, CS) for 8 s followed by the delivery of a food pellet (unconditioned stimulus, US) immediately upon lever-CS retraction. Each PavCA session consisted of 25 lever-food pairings. (**b**) PavCA index scores (composite index of Pavlovian conditioned approach behavior) for individual rats across 5 sessions of Pavlovian conditioning. PavCA index from session 4–5 were averaged to determine the behavioral phenotype. Rats with a PavCA score <−0.3 were classified as goal-trackers (GTs, n = 59), rats with a PavCA score >+0.3 were classified as sign-trackers (STs, n = 56). (**c–e**) Acquisition of lever-directed behaviors (sign-tracking) during PavCA training. Mean ± SEM for (**c**) number of lever contacts, (**d**) probability to contact the lever, and (**e**) latency to contact the lever. (**f–h**) Acquisition of magazine-directed behaviors (goal-tracking) during PavCA training. Mean ± SEM for (**f**) number of magazine entries, (**g**) probability to enter the magazine, and (**h**) latency to enter the magazine. (**i**) Data are expressed as individual data points with mean ± SEM plotted on violin plots for PavCA index. Rats with similar PavCA scores were assigned to receive different G-protein coupled receptor (GPCR; Gi, Gq or no-DREADD) and different treatment (CNO, VEH). Baseline differences in PavCA index between experimental groups were assessed by using a 3-way ANOVA with phenotype (GT, ST), GPCR (Gi, Gq and no-DREADD) and treatment (CNO, VEH) as independent variables and PavCA index as the dependent variable. A significant effect of phenotype was found (p<0.001), but no significant differences between experimental groups and no significant interactions. Sample sizes: GT-Gi = 32, GT-Gq = 12, ST-Gi = 14, ST-Gq = 25, GT-no DREADD = 15, ST-no DREADD = 17.

DOI: https://doi.org/10.7554/eLife.49041.004

The following source data is available for figure 2:

**Source data 1.** Acquisition of lever- and magazine-directed behaviors during 5 sessions of PavCA training (*Figure 2c–h*).
DOI: https://doi.org/10.7554/eLife.49041.005
**Source data 2.** Average PavCA index scores during session 4–5 of PavCA training (*Figure 2b,i*).
DOI: https://doi.org/10.7554/eLife.49041.006

## Inhibition of the PrL-PVT pathway increases the incentive value of the food cue in GTs

Inhibition of the PrL-PVT pathway in GTs significantly increased the PavCA index (*Figure 3f*). This effect appears to be driven primarily by a change in the 'response bias' score ($F_{1,30}$=4.136 p=0.051, Cohens d = 1.04, 1-β=0.99; *data not shown*), which is a measure of: [(total lever-CS contacts − total food magazine entries) / (total lever-CS contacts + total food magazine entries)] (*Meyer et al.,*

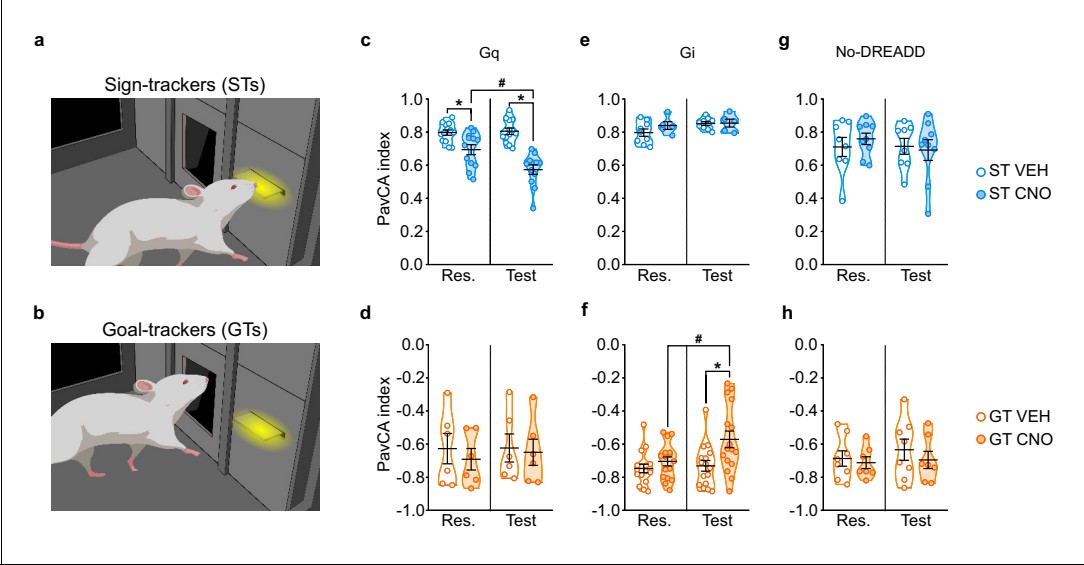

**Figure 3.** Chemogenetic stimulation of the PrL-PVT pathway decreases sign-tracking behavior in sign-trackers, while chemogenetic inhibition of the PrL-PVT pathway increases sign-tracking behavior in goal-trackers. (a,b) Drawing representative of sign-tracking (a) and goal-tracking (b) conditioned responses during PavCA training. (c–h) Data are expressed as individual data points with mean ± SEM plotted on violin plots for PavCA index during Rescreening (Res., average of PCA Sessions 6–10) and Test (average of PCA Sessions 11–16) periods. (c,d) Chemogenetic stimulation (Gq) of the PrL-PVT circuit decreases the PavCA index in (c) sign-trackers, but has no effect in (d) goal-trackers. There was a significant treatment x session interaction in sign-trackers (p<0.01). Pairwise comparisons showed that CNO decreased the PavCA index in STs (*p<0.001, CNO vs. VEH; #p<0.001, Test vs. Res.). (e,f) Chemogenetic inhibition of the PrL-PVT circuit has no effect on (e) sign-trackers, but increases the PavCA index in (f) goal-trackers. There was a significant treatment x session interaction in goal-trackers (*p<0.012 CNO vs. VEH; #p<0.002 Test vs. Res.). (g,h) Effects of CNO administration on the PavCA index in non-DREADD expressing (g) STs and GTs (h). When DREADD receptors were not expressed in the brain, CNO had no effect on the PavCA index in either sign-trackers nor goal-trackers. Sample sizes: GT-Gi = 32, GT-Gq = 12, ST-Gi = 14, ST-Gq = 25, GT-no DREADD = 15, ST-no DREADD = 17.

DOI: https://doi.org/10.7554/eLife.49041.007

The following source data and figure supplements are available for figure 3:

**Source data 1.** Behavioral data during the rescreening and test sessions for ST-Gq (*Figure 3c*).
DOI: https://doi.org/10.7554/eLife.49041.010
**Source data 2.** Behavioral data during the rescreening and test sessions for GT-Gq (*Figure 3d*).
DOI: https://doi.org/10.7554/eLife.49041.011
**Source data 3.** Behavioral data during the rescreening and test sessions for ST-Gi (*Figure 3e*).
DOI: https://doi.org/10.7554/eLife.49041.012
**Source data 4.** Behavioral data during the rescreening and test sessions for GT-Gi (*Figure 3f*).
DOI: https://doi.org/10.7554/eLife.49041.013
**Source data 5.** Behavioral data during the rescreening and test sessions for ST-No DREADD controls (*Figure 3g*).
DOI: https://doi.org/10.7554/eLife.49041.014
**Source data 6.** Behavioral data during the rescreening and test sessions for GT-No DREADD controls (*Figure 3h*).
DOI: https://doi.org/10.7554/eLife.49041.015
**Figure supplement 1.** Analysis of Pavlovian conditioned approach behaviors during the rescreening phase of the study (PavCA sessions 6–10, prior to treatment).
DOI: https://doi.org/10.7554/eLife.49041.008
**Figure supplement 2.** Analysis of magazine entries during the intertrial interval (ITI).
DOI: https://doi.org/10.7554/eLife.49041.009

*2012*). Other specific metrics of lever- or magazine-directed behaviors were not significantly different between treatment groups (*Supplementary file 3* and *4*). For PavCA index, however, there was a significant effect of treatment ($F_{1,30}$=5.975, p=0.021), session ($F_{1,30}$=7.106, p=0.012,), and a significant treatment x session interaction ($F_{1,30}$=4.403, p=0.044, 1-β=0.98). Post-hoc analyses revealed that inhibiting the PrL-PVT pathway increased the PavCA index relative to both the rescreening session (p=0.002, Cohen's d = 0.85) and to VEH controls during the test session (p=0.012, Cohen's d = 0.96). For STs (*Figure 3e*), there was not a significant effect of treatment ($F_{1,12}$=0.876, p=0.368),

session ($F_{1,12}$=4.359, p=0.059), nor a significant treatment x session interaction ($F_{1,12}$=1.479, p=0.247, 1-β=0.85), suggesting that 'turning off' the PrL-PVT pathway permits the attribution of incentive motivational value to a reward-cue selectively in GTs.

## CNO administration in the absence of DREADD receptors does not affect the Pavlovian conditioned approach response in STs or GTs

Administration of CNO in the absence of DREADD had no effect on behavior during PavCA in either STs or GTs (*Figure 3g and h*, respectively).

## Behavior during the intertrial interval was not affected by manipulation of the PrL-PVT pathway or by administration of CNO in the absence of DREADD receptors

To assess the effects of manipulating the PrL-PVT pathway on general locomotor activity and motivated behavior, we examined head entries into the food magazine during the intertrial interval (ITI), when the lever-CS was not present (*Figure 3—figure supplement 2*). Consistent with prior findings, head entries into the food magazine during the ITI tended to decrease with training; thus responses during the 'test' sessions were generally less than those during the 'rescreening' sessions (see statistics in *Figure 3—figure supplement 2* legend). Importantly, however, there were no significant effects of treatment and no significant interactions for this metric for either phenotype or any of the experimental groups (i.e. Gq, Gi, No-DREADD controls). It should also be noted, that all rats continued to consume all of their food pellets during the ITI, regardless of treatment. Thus, the effects described above following manipulation of the PrL-PVT circuit appear to be specific to Pavlovian conditioned approach behavior and not reflective of a change in general locomotor activity or motivated behavior.

## Conditioned reinforcement test

A conditioned reinforcement test (CRT) was conducted to assess the reinforcing properties of the lever-CS (*Robinson and Flagel, 2009*). During this test, responses into a port designated 'active' results in the brief presentation of the lever-CS; whereas those in the 'inactive' port have no consequence. If a rat responds more into the active port relative to the inactive port, the lever-CS is considered to have reinforcing properties (*Robinson and Flagel, 2009*). Moreover, if the rat approaches and interacts with the lever-CS during its brief presentation, it is considered to have incentive properties (*Robinson and Flagel, 2009*; *Hughson et al., 2019*). Here we use the incentive value index, a composite metric ((pokes in active port + lever-CS contacts) – (pokes in inactive port)) as a primary measure of the conditioned reinforcing properties of the lever-CS (*Figure 4*; see also *Hughson et al., 2019*) and additionally report nosepoke responding and lever-CS contacts in *Figure 4—figure supplement 1*.

## Stimulation of the PrL-PVT pathway attenuates the conditioned reinforcing properties of a reward-cue in STs

Stimulation of the PrL-PVT pathway in STs significantly attenuated the incentive value index during the CRT ($t_{25}$ = −3.574, p=0.002, Cohen's d = 1.48, 1-β=0.92, *Figure 4b*). The same manipulation had no effect on the incentive value index in GTs (*Figure 4c*). In agreement, there was a significant effect of treatment ($F_{1,36}$ = 19.021, p<0.001), port ($F_{1,36}$ = 30.501, p<0.001) and a significant treatment x port interaction ($F_{1,36}$ = 7.024, p = 0.012) for nosepoke responding for STs (*Figure 4—figure supplement 1a*). Post-hoc analysis revealed that, relative to VEH controls, stimulation of the PrL-PVT in STs decreased the number of nosepokes into the active port (p<0.001, Cohen's d = 1.55). Furthermore, while VEH-treated controls responded more in the active port relative to the inactive port (p<0.001, Cohen's d = 2.421), this discrimination between ports was abolished following CNO treatment (p=0.061). Similarly, stimulation of the PrL-PVT pathway reduced the number of lever contacts in STs ($t_{18.142}$ = -3.615, p<0.05, Cohen's d = 1.47, *Figure 4—figure supplement 1g*). For GTs, there was not a significant effect of treatment, port, nor a significant treatment x port interaction for nosepoke responding (*Figure 4—figure supplement 1b*), nor a significant effect of treatment for lever-CS interactions (*Figure 4—figure supplement 1h*). These data are consistent with those reported

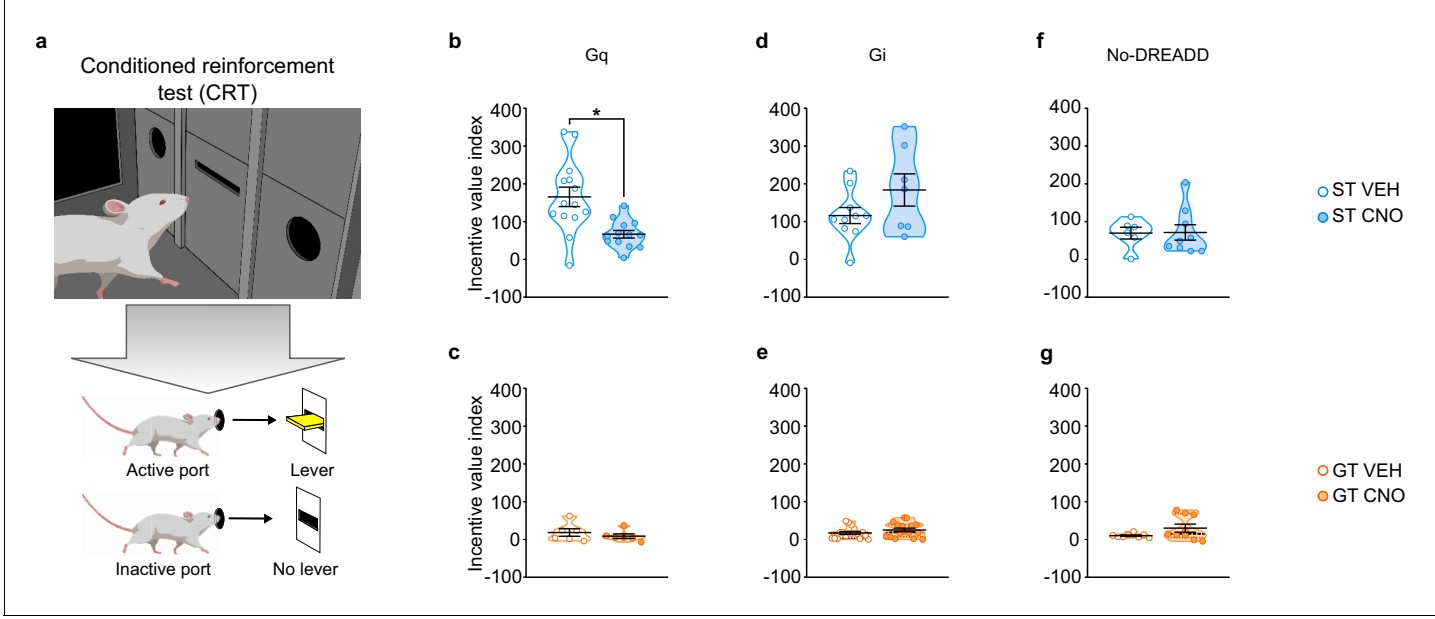

**Figure 4.** Chemogenetic stimulation of the PrL-PVT pathway decreases the conditioned reinforcing properties of a reward-paired cue in sign-trackers. (a) Schematic representing the conditioned reinforcement test (CRT). Data are expressed as individual data points with mean ± SEM plotted on violin plots for Incentive value index ((active nosepokes + lever presses) – (inactive nosepokes)). (b,c) Relative to VEH controls, administration of CNO (3 mg/kp, i.p.) significantly decreased the incentive value of the reward cue for (b) ST-Gq (*p<0.05), but not (c) GT-Gq rats. (d,e) CNO-induced inhibition of the PrL-PVT circuit did not affect the conditioned reinforcing properties of the reward cue in (d) ST-Gi or (e) GT-Gi rats. (f,g) CNO administration did not affect the conditioned reinforcing properties of the reward cue in non-DREADD expressing (f) STs or (g) GTs. Sample sizes: GT-Gi = 32, GT-Gq = 12, ST-Gi = 14, ST-Gq = 25, GT-no DREADD = 15, ST-no DREADD = 17.

DOI: https://doi.org/10.7554/eLife.49041.016

The following source data and figure supplement are available for figure 4:

**Source data 1.** Lever presses (*Figure 4—figure supplement 1h*), nosepokes (*Figure 4—figure supplement 1a*) and incentive value index (*Figure 4b*) during conditioned reinforcement for ST-Gq.

DOI: https://doi.org/10.7554/eLife.49041.018

**Source data 2.** Lever presses (*Figure 4—figure supplement 1h*), nosepokes (*Figure 4—figure supplement 1b*) and incentive value index (*Figure 4c*) during conditioned reinforcement for GT-Gq.

DOI: https://doi.org/10.7554/eLife.49041.019

**Source data 3.** Lever presses (*Figure 4—figure supplement 1i*), nosepokes (*Figure 4—figure supplement 1c*) and incentive value index (*Figure 4d*) during conditioned reinforcement for ST-Gi.

DOI: https://doi.org/10.7554/eLife.49041.020

**Source data 4.** Lever presses (*Figure 4—figure supplement 1j*), nosepokes (*Figure 4—figure supplement 1d*) and incentive value index (*Figure 4e*) during conditioned reinforcement for GT-Gi.

DOI: https://doi.org/10.7554/eLife.49041.021

**Source data 5.** Lever presses (*Figure 4—figure supplement 1k*), nosepokes (*Figure 4—figure supplement 1e*) and incentive value index (*Figure 4f*) during conditioned reinforcement for ST-No DREADD controls.

DOI: https://doi.org/10.7554/eLife.49041.022

**Source data 6.** Lever presses (*Figure 4—figure supplement 1l*), nosepokes (*Figure 4—figure supplement 1f*) and incentive value index (*Figure 4g*) during conditioned reinforcement for GT-No DREADD controls.

DOI: https://doi.org/10.7554/eLife.49041.023

**Figure supplement 1.** Chemogenetic stimulation of the PrL-PVT pathway decreases the conditioned reinforcing properties of a reward-paired cue in sign-trackers.

DOI: https://doi.org/10.7554/eLife.49041.017

above for PavCA behavior, as 'turning on' this top-down cortico-thalamic control attenuates the incentive value of a food-cue selectively in STs.

## Inhibition of the PrL-PVT pathway does not affect the conditioned reinforcing properties of a reward-cue in either STs or GTs

Inhibition of the PrL-PVT pathway did not affect the incentive value index for either phenotype (*Figure 4d and e*). For STs, there was a significant main effect of port ($F_{1,30}$ = 30.696, p<0.001), for which both VEH- and CNO-treated rats responded more in the active port (*Figure 4—figure supplement 1c*). For GTs, there was a significant main effect of treatment ($F_{1,56}$ = 9.608, p = 0.03) and port ($F_{1,56}$ = 23.707, p<0.001), but there was not a significant treatment x port interaction (*Figure 4—figure supplement 1d*). Thus, CNO-treated rats responded more than VEH controls, but they did so across ports (i.e. in both the active and inactive ports); and all GT rats responded more in the active port relative to the inactive port. Inhibition of the PrL-PVT did not affect lever-CS interaction in either phenotype (*Figure 4—figure supplement 1, j*) during the CRT test. Potential explanations for the seemingly incongruent effects of PrL-PVT inhibition on Pavlovian conditioned approach behavior and the conditioned reinforcing properties of the reward cue in GTs are discussed below.

## CNO administration in the absence of DREADD receptors does not affect the conditioned reinforcing properties of a reward-cue in STs or GTs

Administration of CNO in the absence of DREADD did not affect the incentive value index during CRT in either STs (*Figure 4f*) or GTs (*Figure 4g*). For nosepoke responding, there was a significant main effect for port, for which both STs ($F_{1,26}$ = 15.521, p = 0.001) and GTs ($F_{1,26}$ = 8.492, p = 0.007) responded more in the active port relative to the inactive port. There was not a significant effect of treatment, nor a significant treatment x port interaction for either phenotype, suggesting that CNO administration did not affect the conditioned reinforcing properties of the reward cue (*Figure 4—figure supplement 1e–f*). In agreement, CNO administration did not affect lever-CS interaction during the CRT (*Figure 4—figure supplement 1k–l*).

## Experiment 2

### Acquisition of pavlovian conditioned approach

Based on the results of Experiment 1, Experiment 2 focused on STs expressing Gq-DREADD (ST-Gq) and GTs expressing Gi-DREADD (GT-Gi) in neurons of the PrL that project to the PVT (both aPVT and pPVT). For this experiment, the PavCA index from session 3 was used to classify rats as STs and GTs. Since a conditioned response is not fully developed by session 3, different criteria were used for classification. Rats with a PavCA index $\geq$ +0.20 were classified as STs, and rats with an index $\leq -0.20$ were classified as GTs (*Figure 5g*). Intermediate rats with a PavCA index between −0.2 and +0.2 (n = 18) were excluded from the remainder of Experiment 2, and subjects with inaccurate DREADD expression and/or with incorrect microdialysis probe placement (n = 23) were excluded from the statistical analyses, resulting in a final number of 23 rats (GT-Gi = 10, ST-Gq = 13). Analyses of the acquisition of sign- and goal-tracking behaviors across sessions 1–3 of PavCA training is reported in *Supplementary file 5* and *6*.

## Stimulation of the PrL-PVT pathway does not affect the attribution of incentive value to a food cue or the neurochemical profile of the NAcS in STs

Stimulating the PrL-PVT pathway in STs did not affect behavior early in PavCA training (i.e. sessions 3–6; *Figure 5h*, see also *Supplementary file 7* and *8*). These findings suggest that the PrL-PVT pathway mediates the incentive value of a food-cue once the sign-tracking response is acquired, but not during the acquisition process. For microdialysis, there was not a significant effect of treatment (CNO, VEH), time (5 min bins), nor a significant interaction for any of the molecules examined in STs (*Figure 5j*). Noteworthy, however, is a visible trend (*Figure 5k*; p = 0.153) and strong effect size for DA (Cohen's d = 1.26), suggesting that stimulation of the PrL-PVT pathway in STs decreases DA in the NAcS. Yet, consistent with the fact that there were no significant behavioral effects in STs following the same manipulation, there was also not a significant correlation between average DA and PavCA index (*Figure 5l*). The modest sample size (n = 6/7 for behavior, n = 4 per group for microdialysis) may have played a role in the lack of statistical effects in ST-Gq, as suggested by post-hoc power analyses (1-β = 0.21 for behavior; 1-β = 0.28 for microdialysis).

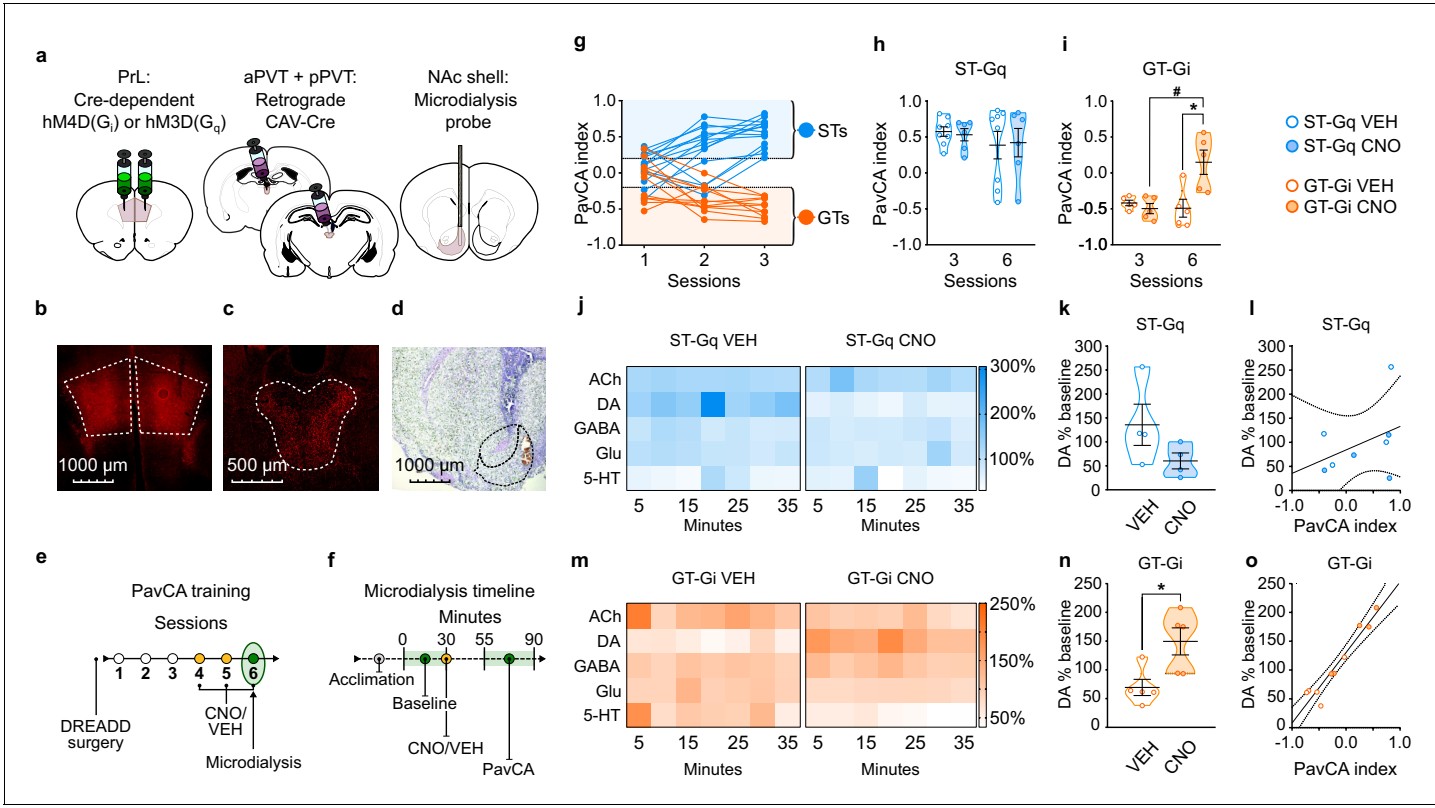

**Figure 5.** Experiment 2: Inhibition of the PrL-PVT circuit early in Pavlovian training elicits sign-tracking behavior and increases extracellular levels of dopamine in the nucleus accumbens shell (NAcS) of GT rats. (a) Schematic of the cannulation surgery and the dual-vector strategy used for expressing Gi-DREADDs in the PrL-PVT pathway of GTs, or Gq-DREADDs in the PrL-PVT pathway of STs. (b–d) Representative pictures of coronal brain slices showing mCherry expression in the (b) PrL and (c) PVT, and the (d) placement of the microdialysis probe in the NAcS. (e) Experimental timeline. After surgery and DREADD incubation, rats were trained in a PavCA task for three consecutive Sessions (1-3) and classified as STs or GTs. Rats then received CNO (3 mg/kg, i.p.) or vehicle (VEH) 25 min before being trained for three additional PavCA Sessions (4-6). (f) Microdialysis timeline. Before the start of Session 6 of PavCA, microdialysis probes were inserted into the guide cannula and rats were left undisturbed for 2 hr before starting sample collection (acclimation period). After acclimation, six baseline dialysates were collected over 30 min before injecting rats with either CNO or vehicle (VEH). Session 6 of PavCA training started 25 min after CNO/VEH injections, and 7 fractions of dialysate were collected during the session. (g) Individual PavCA index scores during the first 3 sessions of Pavlovian conditioning. Rats with a PavCA index <−0.2 were classified as GTs (n = 10), rats with a PavCA index > +0.2 were classified as STs (n = 13). (h,i) Data are expressed as mean ± SEM for PavCA index. (h) CNO during session 6 of training did not affect PavCA index in ST-Gq rats. (i) CNO during session 6 of training significantly increased the PavCA index in GT-Gi rats (*p=0.016 vs. VEH; #p=0.009 vs. session 3). (j,m) Heatmaps showing the relative percent (%) change from baseline of NAc shell acetylcholine (ACh), dopamine (DA), GABA, glutamate (Glu) and serotonin (5-HT) during Session 6 of PavCA training in (j) ST-Gq rats (blue) and (m) GT-Gi rats (orange). (k,n) Mean ± SEM levels of DA % change during Session 6 of Pavlovian conditioning in (k) ST-Gq and (n) GT-Gi rats. There was a significant effect of CNO for GT-Gi rats (*p=0.008). (l,o) correlations between DA % change from baseline and PavCA index during session 6 of PavCA training. No significant correlation was found in (l) ST-Gq. There was a significant positive correlation between percent change in DA and PavCA index in (o) GT-Gi rats ($r^2$ = 0.92; p<0.001). Sample sizes: GT-Gi = 10; ST-Gq = 13 for behavior, nine for microdialysis data.

DOI: https://doi.org/10.7554/eLife.49041.024

The following source data and figure supplement are available for figure 5:

**Source data 1.** PavCA index during session 1–3 of training (*Figure 5g*).
DOI: https://doi.org/10.7554/eLife.49041.026

**Source data 2.** PavCA index during session 3 vs session 6 of training for ST-Gq (*Figure 5h*).
DOI: https://doi.org/10.7554/eLife.49041.027

**Source data 3.** PavCA index during session 3 vs session 6 of training for GT-Gi (*Figure 5i*).
DOI: https://doi.org/10.7554/eLife.49041.028

**Source data 4.** Relative percent change from baseline of NAc shell acetylcholine (ACh), dopamine (DA), GABA, glutamate (Glu) and serotonin (5-HT) during Session 6 of PavCA training in ST-Gq (*Figure 5j*).
DOI: https://doi.org/10.7554/eLife.49041.029

**Source data 5.** Mean levels of NAc shell dopamine % change during Session 6 of PavCA training in ST-Gq (*Figure 5k*).
DOI: https://doi.org/10.7554/eLife.49041.030

*Figure 5 continued on next page*

*Figure 5 continued*

**Source data 6.** Correlations between DA% change and PavCA index during session 6 of training in ST-Gq (*Figure 5l*).
DOI: https://doi.org/10.7554/eLife.49041.031

**Source data 7.** Relative percent change from baseline of NAc shell acetylcholine (ACh), dopamine (DA), GABA, glutamate (Glu) and serotonin (5-HT) during Session 6 of PavCA training in GT-Gi (*Figure 5m*).
DOI: https://doi.org/10.7554/eLife.49041.032

**Source data 8.** Mean levels of NAc shell dopamine % change during Session 6 of PavCA training in GT-Gi (*Figure 5n*).
DOI: https://doi.org/10.7554/eLife.49041.033

**Source data 9.** Correlations between DA% change and PavCA index during session 6 of training in GT-Gi (*Figure 5o*).
DOI: https://doi.org/10.7554/eLife.49041.034

**Figure supplement 1.** Microdialysis probe placement.
DOI: https://doi.org/10.7554/eLife.49041.025

## Inhibition of the PrL-PVT pathway increases the propensity to attribute incentive value to a food cue and DA levels in the NAcS in GTs

Inhibiting the PrL-PVT pathway early in training in GTs resulted in an enhancement of the incentive motivational value of the reward-cue (effect of treatment ($F_{1,8}$ = 10.722, p = 0.011); treatment x session interaction ($F_{1,8}$ = 7.163, p = 0.028; *Figure 5i*). This effect was due, in part, to a decrease in magazine-directed behaviors (*Supplementary file 8*). Post-hoc analysis revealed that CNO treatment significantly increased the PavCA index relative to session 3 (p = 0.016, Cohen's d = 2.40), and relative to VEH-treated rats on session 6 (p<0.05, Cohen's d = 1.94). Despite the small sample size (n = 5 per group) a post-hoc power analysis revealed sufficient power to detect an effect (1-β = 0.95). Thus, inhibition of the PrL-PVT circuit early in training enhances the incentive motivational value of a reward-cue in GTs.

While GABA, Glu, 5-HT and Ach did not differ as a function of treatment or time (*Figure 5m*), DA did (effect of treatment ($F_{1,9.067}$ = 11.145, p = 0.009); treatment x time interaction ($F_{6,50.149}$ = 3.765, p = 0.004)). Relative to VEH-treated rats, CNO treatment resulted in a significant increase in extracellular levels of dopamine across the time-sampling period (*Figure 5m*); and the same pattern was evident when the average levels of DA were compared ($t_8$ = 2.925, p<0.019, Cohen's d = 1.90, 1-β = 0.82, *Figure 5n*). Thus, in GTs, inhibition of the PrL-PVT circuit increases sign-tracking behavior and concomitantly increases DA in the NAcS. Furthermore, there was a significant positive correlation between percent change in DA and PavCA index ($F_{1,9}$ = 91.563, p<0.001; $r^2$ = 0.92), suggesting that DA accounts for more than 90% of the variance in Pavlovian conditioned approach behavior (*Figure 5o*).

## Discussion

The work presented here highlights the PVT as a central node in the integration of top-down and bottom-up circuits involved in the attribution of incentive motivational value to a Pavlovian food-cue. Specifically, we tested the hypothesis that the PrL-PVT circuit serves to suppress the propensity to attribute incentive value to Pavlovian reward-cues, and that the efficacy of this mechanism is dependent on individual differences in cue-motivated behavior. In support, we found that 'turning on' the PrL-PVT pathway via chemogenetic stimulation attenuates the incentive motivational value of a food-cue in sign-tracking rats—those that exhibit a natural propensity to attribute excessive incentive motivational value to such cues—without affecting the behavior of goal-tracking rats. Conversely, 'turning off' the PrL-PVT pathway via chemogenetic inhibition appears to permit the attribution of incentive motivational value to reward cues in GTs, without affecting the behavior of STs.

The current data are consistent with the notion that, in STs, hyper-activation of bottom-up motivational circuits projecting to the PVT override the top-down cognitive information coming from the PrL (*Haight et al., 2017*), thus rendering these individuals more prone to attribute incentive value to a food-paired cue. Here we show that this configuration can be altered with stimulation of the PrL-PVT pathway which, presumably, via the PVT, attenuates subcortical drive and updates the behavioral output, resulting in a reduction in sign-tracking behavior. Such an interpretation also explains the lack of effects observed in STs expressing Gi in the PrL-PVT pathway. In these rats, inhibition of the PrL-PVT pathway is without effect because the relative predominance of bottom-up processes

remains intact. In contrast, in GTs, top-down processes appear to dominate over subcortical processes (*Paolone et al., 2013*; *Haight et al., 2017*; *Koshy Cherian et al., 2017*). As a result, GTs are more goal-driven and more inclined to attribute predictive, but not incentive, value to reward-cues. Thus, inhibition of the PrL-PVT pathway in GTs minimizes the top-down control of the PVT, permitting bottom-up signals to gain a relatively greater influence over their behavior. As a result of this shift in balance, GTs show an increase in the expression of sign-tracking behavior. Correspondingly, stimulation of the PrL-PVT pathway in GTs is without effect, as this manipulation merely reinforces the innate top-down control over behavior in these individuals.

For STs, a discrete food-associated cue is able to elicit approach behavior and support the learning of a new instrumental response, or act as a potent conditioned reinforcer (*Robinson and Flagel, 2009*; *Meyer et al., 2012*). Thus, we wanted to determine if the PrL-PVT pathway plays a role in the ability of a food-cue to act as a conditioned reinforcer (*Berridge et al., 2009b*; *Robinson and Flagel, 2009*). Although different neural mechanisms are known to contribute to Pavlovian conditioned approach behavior and the conditioned reinforcing effects of Pavlovian cues (*Cardinal et al., 2002*), the PrL-PVT pathway appears to play a prominent role in both, at least for STs. That is, in STs, stimulation of the PrL-PVT circuit reduces both sign-tracking behavior and the conditioned reinforcing properties of the reward cue. In GTs, however, inhibition of the PrL-PVT pathway did not affect responding during a test of conditioned reinforcement. Although these findings may seem inconsistent with the increase in sign-tracking behavior following this manipulation in GTs, it is important to note that 'turning off' this top down control did not switch the behavioral phenotype of GTs to STs. In fact, there was only a moderate increase in the PavCA index. Thus, while inactivation of the PrL-PVT pathway permits the attribution of incentive value to the cue to the extent that it elicits sign-tracking behavior in GTs, the ascribed value is not sufficient to support conditioned reinforcement.

As sign-tracking, but not goal-tracking, behavior is dopamine-dependent, we were also interested in determining whether the chemogenetic manipulations that altered behavior would also affect extracellular levels of DA in the NAc. To this end, we assessed the effects of stimulation (in STs) or inhibition (in GTs) of the PrL-PVT pathway on the neurochemical profile of the NAcS, a subregion of the ventral striatum particularly rich in glutamatergic fibers coming from the PVT (*Li and Kirouac, 2008*; *Vertes and Hoover, 2008*). For these studies, the effects of PrL-PVT manipulation were examined early in Pavlovian training, when Pavlovian cues are known to evoke a DA response in the NAc to a different degree in STs and GTs (*Flagel et al., 2011a*; *Clark et al., 2013*). Although we expected to observe more robust behavioral effects during this period, we found that stimulation of the PrL-PVT pathway early in Pavlovian training did not affect the behavior of STs, nor did it significantly affect extracellular levels of DA or other neurotransmitters in the NAcS. In contrast, inhibition of the PrL-PVT pathway early in Pavlovian training increased the tendency to sign-track in GTs, and concomitantly increased extracellular levels of DA in the NAcS, without affecting other neurotransmitters. Thus, in STs, the effects of PrL-PVT stimulation are apparent only after extended Pavlovian conditioning (i.e. Experiment 1); whereas, in GTs, the effects of PrL-PVT inhibition are consistent during early and late stages of Pavlovian conditioning (i.e. in both Experiment 1 and 2).

This pattern of results is likely due to distinct neural mechanisms regulating behavior during 'learning' vs 'performance'. We speculate that the exaggerated contribution of bottom-up processes that characterize STs (e.g. DA release in the NAc) is stronger during the early sessions of training, when the sign-tracking response is still being learned, compared to that after extended training, once the conditioned response has been acquired. Indeed, there is a shift in the neurobiological substrates that mediate the incentive motivational properties of reward cues after prolonged training, such that the dopamine encoding of Pavlovian incentive stimuli eventually diminishes (*Clark et al., 2013*). Therefore, potentiating the top-down communication between the PrL and the PVT during early training may not be strong enough to overcome the subcortical activity surrounding the PVT in STs during this 'critical period' of incentive learning. After learning has taken place, however, subcortical mechanisms may be less critical and, as a consequence of this, sign-tracking behavior may be more malleable, and vulnerable to top-down control. In contrast, in GTs, inhibition of the PrL-PVT circuit increases both the acquisition (learning) and expression (performance) of sign-tracking behaviors, and this effect seems to be even more pronounced early in training. It seems, therefore, that permitting a shift towards bottom-up processing during the acquisition of Pavlovian conditioned approach behaviors results in dopamine-dependent incentive learning, and the PVT appears to act

as a fulcrum in this regard. Further, these data suggest that the predominance of bottom-up processes that contribute to sign-tracking behavior can occur via innate mechanisms (i.e. in STs), or by 'turning off' top-down control mechanisms (i.e. in GTs). Ongoing studies will determine whether chemogenetic or optogenetic manipulations of bottom-up processes can similarly alter individual differences in cue-motivated learning strategies.

It has become increasingly apparent in recent years that the PVT and its associated circuitry play an important role in motivated behaviors, including those related to addiction and anxiety-related disorders (*Coffey et al., 2010*; *Do-Monte et al., 2015*; *Matzeu et al., 2015*; *Zhu et al., 2016*; *Choi and McNally, 2017*; *Do-Monte and Kirouac, 2017b*; *Do-Monte et al., 2017a*; *Matzeu et al., 2017*; *Millan et al., 2017*; *Otis et al., 2017*; *Beas et al., 2018*; *Giannotti et al., 2018*; *Zhu et al., 2018*; *Choi et al., 2019*). Here, we honed in on the PrL-PVT pathway and exploited an animal model of individual differences in cue-reward learning to demonstrate that this circuit acts as a top-down control mechanism to suppress the attribution of incentive value to a food-paired cue. Furthermore, we showed that it does so by affecting the concomitant subcortical processes that instead promote incentive learning, such as DA release in the NAcS. Taken together, these findings identify the PVT as a central node in the integration of top-down and bottom-up mechanisms involved in the attribution of incentive value to reward-cues. The cortico-thalamic-striatal circuitry elucidated here should be considered as a possible target for new therapeutic interventions to prevent the development of, or treat, cue-motivated psychiatric disorders in vulnerable individuals.

## Materials and methods

### Common methods

#### Subjects

A total of 428 male heterogeneous stock (N/NIH-HS) rats from the breeding colony at the Medical College of Wisconsin (LSW, now at Wake Forest School of Medicine) were used for Experiment 1. As we were interested only in assessing STs and GTs, rats with a PavCA index between −0.3 and +0.3 (intermediate rats, n = 182) were excluded from the remainder of Experiment 1. Of the remaining 246 rats, 55 were excluded from the analysis because their phenotype changed between the initial testing and rescreening (27 GTs became STs, 26 GTs and 2 STs became intermediate responders). Sixty-five rats were excluded because of inadequate or off-target DREADD expression in the PrL, aPVT or pPVT, resulting in a final n = 126 rats (GT-Gi = 35 (18 VEH/17 CNO), GT-Gq = 12 (6 VEH/6 CNO), ST-Gi = 17 (10 VEH/7 CNO), ST-Gq = 27 (14 VEH/13 CNO), GT-No DREADD = 17 (8 VEH/9 CNO), ST-No DREADD = 18 (8 VEH/10 CNO)). Data for Experiment 1 were collected across 8 rounds of testing.

Given the results obtained from Experiment 1, Experiment 2 focused on ST rats expressing Gq (ST-Gq) and GT rats expressing Gi (GT-Gi) in the PrL-PVT pathway. A total of 64 male Sprague-Dawley (Charles River Saint-Constant, Canada and Raleigh, NC, USA) rats were used for this study. Intermediate rats with a PavCA index between −0.2 and +0.2 (n = 18) were excluded from the remainder of Experiment 2. Subjects with inadequate or off-target DREADD expression in the PrL, aPVT or pPVT, or with incorrect microdialysis probe placement (n = 23) in the nucleus accumbens shell were excluded from the statistical analysis, resulting in a final number of 23 rats (GT-Gi = 10 (5 VEH/5 CNO), ST-Gq = 13 (7 VEH/6 CNO)). Technical difficulties during sample collection and/or chemical analysis led to the elimination of 5 ST-Gq samples. Thus, the final number of ST-Gq rats for the microdialysis data was 8 (4 VEH, 4 CNO). Animals from Experiment 2 were tested across four different experimental rounds.

All rats were 7–9 weeks old at the time of arrival and were housed in a climate-controlled room (22 ± 2°C) with a 12 hr dark-light cycle (lights on at 06:00 AM or 07:00 AM depending on daylight savings time) and allowed to acclimate to the colony room for at least 1 week prior to handling. All rats from Experiment one had *ad-libitum* access to food and water and were paired-housed for the duration of the experiments. Rats from Experiment two were single-housed after cannula implantation to avoid damage to the implants. Behavioral testing took place during the light cycle between 11:00 AM and 5:00 PM. All procedures followed The Guide for the Care and Use of Laboratory Animals: Eighth Edition (2011, National Academy of Sciences) and were approved by the University of Michigan Institutional Animal Care and Use Committee.

## Viral vectors

Double-floxed $hM_3D(Gq)$ (AAV5-hSyn-DIO-hM$_3$Dq-mCherry, titer of $4.8 \times 10^{12}$ vg/μL) or $hM_4D(Gi)$ (AAV-hSyn-DIO-hM$_4$Di-mCherry, titer of 1.9 to $2.5 \times 10^{13}$ vg/μL) DREADDs were obtained from the University of North Carolina vector core. Canine adenovirus expressing Cre (CAV2-Cre, titer of $\sim 2.5 \times 10^9$ vg/μL) was obtained from Dr. Neumaier at the University of Washington.

## Drugs

Clozapine-N-oxide (CNO) was obtained from the National Institute of Mental Health (NIMH). CNO was dissolved in 6% dimethylsulfoxide (DMSO) in sterile water and administered intraperitoneally (i. p.) at a dose of 3.0 mg/kg. A 6% DMSO solution in sterile water was used as the vehicle (VEH) control.

## Pavlovian conditioned approach (PavCA)

### Apparatus

PavCA training occurred inside Med Associates chambers (St. Albans, VT, USA; $30.5 \times 24.1 \times 21$ cm) located in sound-attenuating cabinets with a ventilation fan to mask background noise. Each chamber contained a food magazine located in the center of one wall located 3 cm above the grid floor, connected to an automatic pellet dispenser. A retractable backlit metal lever was located either to the left or right of the food magazine, 6 cm above the grid floor. A house light was located on the wall opposite to the food magazine and lever, 1 cm from the top of the chamber. Magazine entries were recorded upon break of a photo-beam located inside the magazine and a lever contact was registered upon deflection, which required a minimum of a 10-g force.

### PavCA procedures

PavCA procedures were the same as those described previously (*Meyer et al., 2012*). For two days before the start of behavioral procedures, rats were handled by the experimenters and given ~25 banana-flavored food pellets (45 mg dustless pellets, Bio-Serv, Flemington, NJ, USA) in order to familiarize them with the food reward that served as the US during PavCA training. After this period, rats underwent two pre-training sessions, consisting of 25 trials in which a food pellet was delivered into the food magazine on a variable time (VT) 30 s schedule (range 0–60 s). During pre-training, head entries into the food magazine were recorded to ensure that rats retrieved all of the food pellets. Each session lasted approximatively 12.5 min. After pre-training, rats underwent one daily session of PavCA training for five consecutive days. Each PavCA training session consisted of 25 trials under a VT-90 s schedule (range 30–150 s), during which the presentation of an illuminated lever-CS for 8 s was followed by the delivery of a banana-flavored food pellet into the food magazine. Thus, each trial of lever-CS/food-US pairing occurred, on average, every 90 s, within an intertrial interval range of 30–150 s. The start of PavCA training was signaled by illumination of the house light and lasted an average of 40 min. For each PavCA session, the number of lever-CS contacts and head entries into the food magazine, the probability of contacting the lever-CS or entering the food magazine, and the latency to contact the lever-CS or to enter the food magazine during each trial were recorded. These measures were used to calculate a PavCA index as previously described (*Meyer et al., 2012*). Briefly, the PavCA index is a composite score that captures the propensity to approach the lever-CS relative to the food magazine. It is calculated by averaging three distinct measures: 1) response bias [(total lever-CS contacts − total food magazine entries) / (total lever-CS contacts + total food magazine entries)], 2) probability difference [Prob$_{(lever)}$ − Prob$_{(food\ magazine)}$] and, 3) latency difference [−(latency to contact the lever-CS − latency to enter the food magazine)/ 8]. The PavCA index ranges from +1 to −1. A score of −1 indicates an extreme GT whose conditioned responses are always directed toward the food magazine, while a score of +1 indicates an extreme ST whose conditioned responses are always directed toward the lever-CS. For Experiment 1, rats with an average PavCA index $\geq$ +0.30 during session 4 and 5 were classified as STs, while rats with an index $\leq$−0.30 were classified as GTs. For Experiment 2, rats with an average PavCA index $\geq$ +0.20 during session three were classified as STs, while rats with an index $\leq$−0.20 were classified as GTs.

## Surgery

For both Experiments 1 and 2, a dual-vector approach was used to selectively express DREADD receptors in neurons of the PrL that project to the aPVT and pPVT. Rats were deeply anesthetized using 5% inhaled isoflurane (Isothesia - Butler-Schein, Columbus, OH) and the anesthetic plane was maintained with 2% inhaled isoflurane for the duration of the surgery. Prior to surgeries, while under anesthesia, rats received an injection of carprofen (5 mg/kg, s.c.) for analgesia and were further prepared for surgeries by shaving the scalp and applying betadine (Purdue Products, Stamford, CT) followed by 70% alcohol as an antiseptic. Rats were then placed into a stereotaxic frame (David Kopf instruments, Tujunga, CA or Stoelting, Wood Dale, IL) and a small incision was made on the scalp to expose the skull. Two small holes were drilled above the PrL (bilaterally) and above the anterior PVT (aPVT) and posterior PVT (pPVT) using Bregma coordinates as indicated below. Cre-dependent DREADD viruses (Gq or Gi) were bilaterally injected into the PrL (from bregma: +3.0 mm AP;±1.0 mm ML; −4 mm DV) at a rate of 200 nL per minute over a 5-min period (1 µL total). CAV2-Cre was injected at a 10° angle into the aPVT (from bregma: −2.0 mm AP; −1 mm ML; −5.4 mm DV) and pPVT (from bregma: −3.0 mm AP; −1 mm ML; −5.5 mm DV) at a rate of 50 nL per minute for 2 min (100 nL total volume). Following delivery, injectors were left in place for an additional 5 min to allow diffusion. For Experiment 2, rats were additionally implanted with a guide cannula (CMA 12, Harvard Apparatus, Holliston, MA), counterbalanced in the left or right NAcS (from bregma: +1.3 mm AP;±0.8 mm ML; −5.2 mm DV). Cannulas were secured to the skull with stainless steel screws and acrylic cement (Bosworth New Truliner, Keystone Industries, Gibbstown, NJ). Removable stylets were placed in the guide cannulas to avoid clogging. Behavioral tests were conducted 3–5 weeks following surgery.

## Perfusion and tissue processing

Approximately 1 week after completing the behavioral experiments, rats were anesthetized with ketamine (90 mg/kg i.p.) and xylazine (10 mg/kg i.p.) and transcardially perfused with 0.9% saline followed by 4% formaldehyde (pH = 7.4) for fixation. Brains were extracted, post-fixed in 4% formaldehyde for 24 hr at 4°C and immersed in increasing concentrations of sucrose solutions every 24 hr (10%, 20% then 30% sucrose in 0.1M PBs, pH = 7.4) at 4°C over the course of 3 days. Brains were then encased in Tissue-Plus O.C.T. (Fisher HealthCare, Houston, TX), frozen using dry ice and subsequently sectioned in the coronal plane (40 µm) using a cryostat (Leica Biosystems Inc, Buffalo Grove, IL). For Experiment 1, brain slices from the PrL (ranging from +5.00 to +2.5 mm AP, relative to Bregma) and PVT (ranging from −1.00 to −4.00 mm AP, relative to Bregma) were placed into well plates containing cryoprotectant and stored at −20°C until further processing.

## Immunohistochemistry

The accuracy of DREADD expression in the PrL and PVT was assessed using immunohistochemical staining methods to visualize the mCherry protein in DREADD-expressing neurons. Free-floating coronal sections from the PrL and PVT were first rinsed 3 times in 0.1M PBS (pH = 7.4). Endogenous peroxidase activity was blocked by incubating sections in 1% $H_2O_2$ for 10 min, followed by three additional rinses. To prevent non-specific binding of the secondary antibody, sections were incubated in 0.1M PBS containing 0.4% Triton X-100 (TX) and 2.5% Normal Donkey Serum (NDS) (Jackson ImmunoResearch Laboratories, Inc, West Grove, PA). Sections were then incubated overnight at room temperature in primary antibody (rabbit anti-mCherry, Abcam, Cambridge, UK, diluted 1: 30,000) in 0.1M PBS + 0.4% TX + 1% NDS. On the following day, sections were rinsed again before being incubated for 1 hr in a biotinylated donkey anti-rabbit secondary antibody (Jackson Immunoresearch, West Grove, PA, diluted 1: 500) in 0.1M PBS + 0.4% TX + 1% NDS. Peroxidase staining was obtained with a standard avidin-biotin procedure using the Vectastain Elite ABC Kit (Vector Laboratories, Inc, Burlingame, CA diluted 1: 1000 for A and B). Chromogenic reaction occurred by incubating sections in a 0.1M PBS solution containing 0.02% 3,3'-diaminobenzidine tetrahydrochloride (DAB) and 0.012% $H_2O_2$. Sections were rinsed and stored at 4°C until mounted, air dried and coverslipped with Permount (Thermo-Fisher Scientific, Waltham, MA). Bright-field images containing the PrL, aPVT and pPVT were captured using a Leica DM1000 light microscope (Leica-Microsystems, Wetzlar, GER) and were analyzed by two experimenters blind to the experimental groups. Experimenters assigned a score of 0–3 to each image according to both the intensity and location of

DREADD expression in the areas of interest (*Figure 1—figure supplement 1*). A score of 0 was assigned to subjects that had no DREADD expression or off target DREADD expression (e.g. expression outside target boundaries) in either the PrL or the aPVT and pPVT; a score of 1 was assigned to subjects that had adequate DREADD expression in both the PrL and PVT; a score of 2 was assigned to subjects that had a strong DREADD expression in either the PrL or PVT; a score of 3 was assigned to subjects that had a strong DREADD expression in both the PrL and PVT. Representative images of what was considered adequate DREADD expression are shown in *Figure 1—figure supplement 1*. Rats that had a score of 0 (n = 65) were excluded from the statistical analysis.

For representative purposes, in a subset of brains DREADD expression in the PrL and PVT was assessed using immunofluorescence to visualize mCherry in DREADD-expressing neurons. Sections were incubated overnight at room temperature in primary antibody (rabbit anti-mCherry, Abcam, Cambridge, UK, diluted 1: 500) in 0.1M PBS + 0.4% TX + 2.5% NDS. On the following day, sections were rinsed and incubated for 2 hr in a biotinylated donkey anti-rabbit secondary antibody (Jackson Immunoresearch, West Grove, PA, diluted 1: 500) in 0.1M PBS + 0.4% TX + 2.5% NDS. Sections were rinsed again and then incubated for 1 hr in Alexa Fluor 594-conjugated streptavidin (Thermo Fisher Scientific, Waltham, MA, diluted 1:1000) in 0.1M PBS + 0.4% TX. Sections were then mounted onto slides and cover-slipped with ProLong Gold Antifade Mountant (Thermo Fisher Scientific, Waltham, MA). Fluorescent pictures containing the PrL, aPVT and pPVT were captured using a Zeiss AxioImager M2 motorized fluorescent microscope with Apotome structured illumination (Carl Zeiss, Sweden).

## Statistical analyses

For Experiment 1, acquisition of sign- and goal-tracking behavior during sessions 1–5 of PavCA training was analyzed using the following dependent variables: number of lever contacts, number of magazine entries during the CS period, probability to contact the lever, probability to enter the magazine during the CS period, latency to contact the lever and latency to enter the magazine during the CS period. To assess differences in the acquisition of sign- and goal-tracking behaviors during PavCA training, a Linear Mixed-effects Model (LMM) with a Restricted Maximum Likelihood (REML) estimation method was used. Session was used as the repeated variable and GPCR (three levels: Gq, Gi and No-DREADD), phenotype (two levels: GT, ST) and treatment (two levels: CNO, VEH) as between-subject variables. Before choosing the final model, all covariance structures were explored for each one of the dependent variables and the best-fitting model was chosen by selecting the lowest Akaike Information criterion (AIC) (*Greet and Greet, 2009*; *Duricki et al., 2016*). To ensure that subjects were counterbalanced between different experimental groups after PavCA training, the average PavCA index from session 4 and 5 was analyzed by using a 3-way ANOVA with phenotype (GT, ST), GPCR (Gq, Gi and No-DREADD) and treatment (CNO, VEH) as independent variables. It should be noted that comparisons between phenotypes and treatment groups were only made for the acquisition phase of PavCA behavior, as it is important to demonstrate that the phenotypes differ in their conditioned response and that the classification and assignment to GPCR (Gq, Gi, No-DREADD) and treatment (CNO, VEH) groups was balanced within phenotype according to these measures.

To ensure that sign- and goal-tracking conditioned responses were stable after DREADD incubation, we analyzed the PavCA index during each daily session of rescreening (session 6–10) for each GPCR and phenotype using a LMM with REML as the estimation method. Session (five levels: session 6,7,8,9,10) was used as the repeated measure and treatment (two levels: CNO, VEH) as the between subject variable.

To test the effects of either stimulation (i.e. Gq-expressing rats) or inhibition (i.e. Gi-expressing rats) of the PrL-PVT pathway on the expression of a Pavlovian conditioned approach response, we compared the average PavCA index during PavCA rescreening (sessions 6–10, prior to treatment) with the average PavCA index during test sessions (11–16, concurrent with treatment). Thus, the PavCA index from sessions 6–10 of PavCA rescreening and from the PavCA test (sessions 11–16) were averaged and compared for each GPCR and phenotype using a LMM with REML as the estimation method. Session (two levels: rescreening, test) was used as the repeated measure and treatment (two levels: CNO, VEH) as the between-subject variable.

To test the specificity of the effects of the stimulation or inhibition of the PrL-PVT pathway on Pavlovian conditioned approach behaviors, we analyzed the magazine entries during the intertrial interval (ITI), which can be used as an index of general locomotor activity (*Campus et al., 2016*; *Fraser et al., 2016*). For each GPCR and Phenotype we used a LMM with REML as the estimation method to assess the effect of the manipulation of the PRL-PVT pathway on the average ITI magazine entries during PavCA rescreening (sessions 6–10, prior to treatment) with the average ITI magazine entries during test (sessions 11–16, concurrent with treatment). Session (two levels: rescreening, test) was used as the repeated measure and treatment (two levels: CNO, VEH) as the between-subject variable.

To analyze the effect of CNO administration on the behaviors expressed during the CRT test, nose pokes, lever contacts and incentive value index during CRT were analyzed separately for each GPCR and phenotype. Nose pokes were analyzed using a two-way ANOVA with nose port (two levels: active, inactive) and treatment (two levels: CNO, VEH) as independent variables. Differences between CNO- and VEH-treated animals for lever contacts and the incentive value index were analyzed using an unpaired t-test.

For Experiment 2, acquisition of sign- and goal-tracking behavior during sessions 1–3 of PavCA training was analyzed using the same dependent variables described above: number of lever contacts, number of magazine entries during the CS period, probability to contact the lever, probability to enter the magazine during the CS period, latency to contact the lever and latency to enter the magazine during the CS period. To assess differences in the acquisition of sign- and goal-tracking behaviors during PavCA training a LMM with a REML estimation method was used. Session was used as the repeated variable and group (two levels: GT-Gi, ST-Gq) and treatment (two levels: CNO, VEH) as between-subject variables. The best fitting model was chosen by selecting the lowest AIC. To explore the existence of baseline differences between different experimental groups after PavCA training, the PavCA index from session three was analyzed by using a 2-way ANOVA with group (GT-Gi, ST-Gq), and treatment (CNO, VEH) as independent variables. For the same reasons as those described above for Experiment 1, this is the only analysis for which the two groups were directly compared. To analyze the effects of CNO administration within each group, the PavCA index expressed after the initial acquisition (Session 3) was compared with the PavCA index expressed after 3 days of treatment (Session 6). Session (3 vs. 6) was used as the repeated measure and treatment (CNO, VEH) as the between-subject variable within each group. Data were analyzed using a LMM with REML as estimation method.

Microdialysis data were also analyzed separately for ST-Gq and GT-Gi rats. The following neurotransmitters were analyzed: gamma-Aminobutyric acid (GABA), glutamate (Glu), dopamine (DA), serotonin (5-HT) and acetylcholine (ACh). For all neurotransmitters analyzed, differences between treatment groups in the baseline levels were analyzed using an unpaired sample t-test. Levels of neurotransmitters obtained during the subsequent block of collection (pre-PavCA and PavCA) were expressed in terms of percentage (%) change from baseline. Differences in the % change from baseline for each neurotransmitter were analyzed by using a LMM with a REML estimation method. Time was used as a repeated measure and treatment (CNO, VEH) as a between-subject independent variable. Differences in the averaged % change from baseline during PavCA training were analyzed using a t-test. The relationship between DA and PavCA index during session 6 of PavCA training was examined using a linear regression with % change of DA as the predictor and PavCA index as the dependent variable. A one-way ANOVA with PavCA index as the dependent variable and treatment (VEH, CNO) as the independent variable, with% DA as a covariate, was used to determine if the relationship between DA and PavCA index changed as a function of treatment.

For all LMM and ANOVA analyses, normality was checked using a Shapiro-Wilk test. If normality was violated, the presence of outliers and influential data points was explored using boxplots (*Tukey, 1977*). Data points that fell below −1.5 or above +1.5 times the interquartile range (IQR) were considered as outliers and excluded from the statistical analysis in order to meet the assumption of normality. Using these criteria, 11 subjects were excluded from Experiment 1 (GT-Gq = 2; ST-Gi = 3; GT-Gi = 3; ST-No DREADD = 1; GT-No DREADD = 2), resulting in an n = 115 for the PavCA index analyses. Importantly, however, the significant results of the statistical analyses were the same with or without the outliers. Graphical representations and tables include only those data without outliers. All statistical analyses were performed using IBM SPSS Statistics 24 (IBM, Armonk, NY, USA). Alpha was set at 0.05. When significant main effects or interactions were detected, Bonferroni

post-hoc comparisons were performed. Effects sizes for pairwise comparisons were calculated using Cohen's d (*Cohen, 1988*), as previously reported (*Hughson et al., 2019*). Effect sizes < 0.2 were considered small, effects sizes $\geq 0.2 \leq 0.8$ were considered medium, and effect sizes > 0.8 were considered large (*Cohen, 1988*; *Sawilowsky, 2009*). Optimal sample sizes (n) were determined a-priori based on previous publications (*Haight et al., 2017*; *Kuhn et al., 2018*). However, because it is impossible to predict how many rats will be classified as STs or GTs, and how many animals will be excluded after the histological analysis, the final sample sizes varied considerably among experimental groups (see Subjects section). Given the variability in sample sizes, the power to correctly reject the null hypothesis (1-β) was calculated by post-hoc power analyses using G*Power (*Faul et al., 2007*; *Faul et al., 2009*). The threshold for statistical power to be considered sufficient was set at 0.80 (*Cohen, 1992*). For illustration of group comparisons, individual data points are plotted with mean and SEM overlaid on a violin plot to show the distribution of the dependent variable. For statistical analysis with multiple factors, representation of main effects is omitted, and symbols indicate significant pairwise comparisons following a significant interaction. All graphical representations of the data were made using Prism 8 (Graphpad Software, San Diego, CA). Violin plots and individual data points with mean ± SEM were overlaid using Inkscape 0.92 for Windows.

## Experiment 1 detailed methods

### PavCA rescreening and test

The experimental design for Experiment one is summarized in *Figure 1a*. Following recovery from surgery and DREADD incubation, rats underwent 5 PavCA rescreening sessions (sessions 6–10) to determine if their initial behavioral characterization as a ST or GT had changed. To habituate rats to the injection procedures, all rats received an injection (i.p.) of vehicle 25 min prior to session 10. The PavCA testing phase occurred during sessions 11–16, and rats received administration of CNO or VEH 25 min prior to each session. For all analyses, session (rescreening vs. test) was used as the within-subject independent variable and treatment (CNO, VEH) as the between-subject independent variable. PavCA index is presented as the primary dependent variable in the main text, but analyses for other dependent variables are included in the Supplementary files.

### Conditioned reinforcement test

The day after the completion of PavCA training, rats were exposed to a conditioned reinforcement test (CRT) as described previously (*Hughson et al., 2019*). The CRT can be used for assessing the ability of incentive conditioned stimuli to act as a reinforcer and support the learning of a new instrumental response in the absence of a primary reinforcer (*Robinson and Flagel, 2009*). Briefly, for CRT conditioning chambers were reconfigured such that the food magazine was removed, and the retractable lever was placed in the center of the wall flanked by a nose-port on either side. Nose pokes into one port (active port), resulted in the presentation of the illuminated lever for 2 s on a fixed-ratio 1 (FR1) schedule; nose pokes into the opposite port (inactive port) had no consequence. To minimize side bias, the active port was always placed opposite the side the lever-CS was located during PavCA sessions. The conditioned reinforcement test lasted 40 min, and the following behaviors were recorded: number of lever contacts, number of nose pokes into the active port (active nose pokes) and number of nose pokes into the inactive port (inactive nose pokes). A composite score, the incentive value index (*Hughson et al., 2019*), was then used to capture all of the behavioral measures obtained during the conditioned reinforcement test: ((active nose pokes + lever presses) – (inactive nose pokes)).

## Experiment 2 detailed methods

### Drugs

All chemicals, drugs, and reagents were purchased from Sigma-Aldrich unless otherwise noted. HPLC grade water was purchased from Thermo-Fisher Scientific (Waltham, MA) and HPLC grade acetonitrile was purchased from VWR (Radnor, PA).

### PavCA training

The experimental design for Experiment 2 is summarized in *Figure 5a*. PavCA training apparatus and experimental procedures were the same as described above. However, during all PavCA

training sessions rats were tethered to a swiveled arm to habituate them to the microdialysis procedures. The tethered rats could explore the chamber freely. In this experiment, rats were phenotyped as ST or GT following just three initial PavCA sessions. Since the sign- and goal-tracking conditioned responses are not fully developed by session 3, we used a different cut-off for classifying rats as STs or GTs. For this study rats with a PavCA index $\geq$ +0.20 were classified as STs and rats with a score $\leq -0.20$ were classified as GTs. Rats received CNO or VEH 25 min prior to the PavCA training sessions 4–6.

## Microdialysis

Microdialysis methods are depicted in *Figure 5e*. Extracellular levels of neurotransmitters in the NAcS were assessed during session 6 of PavCA training. The percentage change from baseline was measured for the following neurotransmitters: GABA, glutatmate (Glu), dopamine (DA), serotonin (5-HT) and acetylcholine (Ach). Briefly, before session 6 of PavCA training, microdialysis probes were inserted in the NAcS and perfused at a rate of 1.0 µl/min with artificial cerebrospinal fluid (aCSF, pH 7.4) containing, in mM: 145.00 NaCl, 2.68 KCl, 1.10 MgSO4, 1.22 CaCl2, 0.50 NaH2PO4, 1.55 Na2HPO4. Immediately before starting perfusion aCSF was added with 0.25 mM ascorbic acid to prevent dopamine oxidation. Dialysate samples (5 µl) were collected every 5 min, beginning 120 min after probe insertion. Baseline samples were collected for 30 min (six fractions). After this time rats were injected with CNO or VEH and samples were collected for an additional 25 min (five fractions). PavCA training started 25 min after CNO or VEH injections. Samples were collected during the PavCA session for a total time of 35 min (PavCA samples, seven fractions). After collection, samples were derivatized as described below and stored at −80°C until they were analyzed by high-performance liquid chromatography-tandem mass spectrometry (HPLC-MS).

## Benzoyl chloride derivatization

Benzoyl chloride derivatization of dialysates and internal standards was performed using a modified version of the methods described in *Song et al. (2012)*. Calibration curves were generated using standards at 0, 0.1, 0.5, 1, 5, 10, and 20 nM for DA. Internal standards (IS) stock solutions were derivatized by adding 100 mM sodium carbonate monohydrate buffer, followed by 2% $^{13}C_6$ benzoyl chloride in acetonitrile with 0.1% formic acid. The IS stock was then diluted 100-fold in a 50:50 acetonitrile/$H_2O$ solution containing 1% sulfuric acid. Calibration standards and dialysate samples were derivatized by adding them with 100 mM sodium carbonate buffer, 2% benzoyl chloride (in acetonitrile), and an internal standard in a 2:1:1:1 ratio.

## Hplc-ms

A Thermo Finnigan Surveyor Plus HPLC system was used for analyzing microdialysis samples. Neurochemical separation was achieved with a Phenomenex (Torrance, CA) Kinetex biphenyl LC column (50 × 2.1 mm, 1.7 µm particle size, 100 Å pore size). Mobile phase A was 10 mM ammonium formate and 0.15% (v/v) formic acid in HPLC water. Mobile phase B was acetonitrile. The mobile phase gradient for all of the analytes was: initial, 0% B; 0.1 min, 10% B; 0.12 min, 10% B; 2.3 min, 20% B; 3.7 min, 50% B; 4.0 min, 80% B; 4.5 min, 100% B; 5.0 min 100% B; 6.5 min, 0% B. The flow rate was 200 µL/min, and the sample injection volume was 7 µL. The autosampler and column were maintained at ambient temperature throughout the analysis. A Thermo Finnigan TSQ Quantum Ultra triple quadrupole mass spectrometer operating in positive mode was used for detection. Electrospray ionization (ESI) voltage was 3.5 kV, and heated ESI probe (HESI-I) was set at 300°C. Capillary temperature was 350°C, and sheath gas, aux gas, and ion sweep gas were maintained at 25, 15, and 0 arb, respectively. The intercycle delay was 200 ms. Automated peak integration was performed using Thermo X Calibur Quan Browser version 2.1. All peaks were visually inspected to ensure proper integration. Calibration curves were constructed based on peak area ratio ($P_{analyte}/P_{IS}$) versus concentrations by linear regression.

## Immunohistochemistry and histology

Upon completion of experimental procedures, brain tissue was processed for the assessment of DREADD expression as described above. To identify the placement of the microdialysis probes, brain slices containing the NAc shell (bregma 1.70–1.00 mm AP) were mounted onto glass slides,

stained using Cresyl-violet (Sigma-Aldrich, St. Louis, MO) and cover-slipped with a toluene-based solution (Permount, Fisher Scientific, Fair Lawns, NJ). Verification of probe placement was done using a Leica DM1000 light microscope (Buffalo Grove, IL) by two experimenters blind to experimental groups. Only rats with probe placements within the NAc shell were included in the statistical analyses.

## Acknowledgements

Funding for this work was provided by the National Institute on Drug Abuse (NIDA) branch of the National Institutes of Health: R01-DA038599 (SBF), T32-DA007821 (BNK, SAL) and T32-DA007268 (IRC, AP).

We would like to acknowledge the technical assistance of Katie Long, Marin Klumpner and Maurice Chojecki throughout the course of these experiments, as well as assistance from Zhongyan Gong. In addition, we would like to thank Katie Holl for maintenance of the heterogenous stock rat colony at the Medical College of Wisconsin, and Matthew Folz for assistance with confocal imaging in the Molecular and Behavioral Neuroscience Institute at the University of Michigan. We would also like to acknowledge the Flagel Lab members who helped build the foundation for this work, including Dr. Joshua Haight and Kurt Fraser.

## Additional information

### Funding

| Funder | Grant reference number | Author |
|---|---|---|
| National Institute on Drug Abuse | R01-DA038599 | Shelly B Flagel |
| National Institute on Drug Abuse | T32-DA007268 | Ignacio R Covelo<br>Aram Parsegian |
| National Institute on Drug Abuse | T32-DA007821 | Brittany N Kuhn<br>Sofia A Lopez |

The funders had no role in study design, data collection and interpretation, or the decision to submit the work for publication.

### Author contributions

Paolo Campus, Data curation, Software, Formal analysis, Investigation, Visualization, Methodology, Writing—original draft, Writing—review and editing; Ignacio R Covelo, Conceptualization, Investigation, Methodology, Writing—review and editing; Youngsoo Kim, Formal analysis, Investigation, Writing—review and editing; Aram Parsegian, Software, Visualization, Methodology, Writing—review and editing; Brittany N Kuhn, Sofia A Lopez, Investigation, Writing—review and editing; John F Neumaier, Susan M Ferguson, Leah C Solberg Woods, Resources, Writing—review and editing; Martin Sarter, Conceptualization, Writing—review and editing; Shelly B Flagel, Conceptualization, Resources, Supervision, Funding acquisition, Writing—review and editing

### Author ORCIDs

Paolo Campus  https://orcid.org/0000-0002-6513-6158
Ignacio R Covelo  https://orcid.org/0000-0003-2469-2454
Shelly B Flagel  https://orcid.org/0000-0002-7309-9908

### Ethics

Animal experimentation: All animals were handled according to The Guide for the Care and Use of Laboratory Animals: Eighth Edition (2011, National Academy of Sciences) and were approved by the University of Michigan Institutional Animal Care and Use Committee.

**Decision letter and Author response**
Decision letter https://doi.org/10.7554/eLife.49041.045
Author response https://doi.org/10.7554/eLife.49041.046

## Additional files

### Supplementary files

• Supplementary file 1. Acquisition of Pavlovian conditioned approach during PavCA Sessions 1–5: lever-directed behaviors. The results of linear mixed model analyses are shown for the effect of treatment (VEH vs. CNO) across sessions 1–5 of Pavlovian conditioned approach (PavCA) training for lever-directed behaviors, (lever contacts, probability to contact the lever and latency to contact the lever). Analyses were conducted separately for each experimental group (ST-Gq, GT-Gq, ST-Gi, GT-Gi, ST-no DREADD, GT-no DREADD). Bolded values indicate statistical significance, $p < 0.05$.
DOI: https://doi.org/10.7554/eLife.49041.035

• Supplementary file 2. Acquisition of Pavlovian conditioned approach during PavCA Sessions 1–5: magazine-directed behaviors. The results of linear mixed model analyses are shown for the effect of treatment (VEH vs. CNO) across sessions 1–5 of Pavlovian conditioned approach (PavCA) training for magazine-directed behaviors (magazine entries, probability to enter the magazine, and latency to enter the magazine). Analyses were conducted separately for each experimental group (ST-Gq, GT-Gq, ST-Gi, GT-Gi, ST-no DREADD, GT-no DREADD). Bolded values indicate statistical significance, $p < 0.05$.
DOI: https://doi.org/10.7554/eLife.49041.036

• Supplementary file 3. PavCA rescreening (Sessions 6–10) vs. PavCA test (Sessions 11–16): lever-directed behaviors. The results of linear mixed model analyses are shown for the effect of treatment (VEH vs. CNO), session (rescreening vs, test) and treatment x session interaction for lever-directed behaviors (lever contacts, probability to contact the lever and latency to contact the lever). Analyses were conducted separately for each experimental group (ST-Gq, GT-Gq, ST-Gi, GT-Gi, ST-no DREADD, GT-no DREADD). Bolded values indicate statistical significance, $p < 0.05$.
DOI: https://doi.org/10.7554/eLife.49041.037

• Supplementary file 4. PavCA rescreening (sessions 6–10) vs. PavCA test (sessions 11–16): magazine-directed behaviors. The results of linear mixed model analyses are shown for the effect of treatment (VEH vs. CNO), session (rescreening vs, test) and treatment x session interaction for magazine-directed behaviors (magazine entries, probability to enter the magazine and latency to enter the magazine). Analyses were conducted separately for each experimental group (ST-Gq, GT-Gq, ST-Gi, GT-Gi, ST-no DREADD, GT-no DREADD). Bolded values indicate statistical significance, $p < 0.05$.
DOI: https://doi.org/10.7554/eLife.49041.038

• Supplementary file 5. Acquisition of sign-tracking behavior during PavCA Sessions 1–3: lever-directed behaviors. The results of linear mixed model analyses are shown for the effect of treatment (VEH vs. CNO) across sessions 1–3 of Pavlovian conditioned approach (PavCA) training for lever-directed behaviors (lever contacts, probability to contact the lever and latency to contact the lever). Analyses were conducted separately for each experimental group (ST-Gq, GT-Gi). Bolded values indicate statistical significance, $p < 0.05$.
DOI: https://doi.org/10.7554/eLife.49041.039

• Supplementary file 6. Acquisition of sign-tracking behavior during PavCA Sessions 1–3: magazine-directed behaviors. The results of linear mixed model analyses are shown for the effect of treatment (VEH vs. CNO) across sessions 1–3 of Pavlovian conditioned approach (PavCA) training for magazine-directed behaviors (magazine entries, probability to enter the magazine and latency to enter the magazine). Analyses were conducted separately for each experimental group (ST-Gq, GT-Gi). Bolded values indicate statistical significance, $p < 0.05$.
DOI: https://doi.org/10.7554/eLife.49041.040

• Supplementary file 7. Session 3 vs. Session 6 of PavCA training: lever-directed behaviors. The results of linear mixed model analyses are shown for the effect of treatment (VEH vs. CNO), session (3 vs. 6) and treatment x session interaction for lever-directed behaviors (lever contacts, probability

to contact the lever and latency to contact the lever). Analyses were conducted separately for each experimental group (ST-Gq, GT-Gi). Bolded values indicate statistical significance, p<0.05.
DOI: https://doi.org/10.7554/eLife.49041.041

• Supplementary file 8. Session 3 vs. Session 6 of PavCA training: magazine-directed behaviors. The results of linear mixed model analyses are shown for the effect of treatment (VEH vs. CNO), session (3 vs. 6) and treatment x session interaction for magazine-directed behaviors (magazine entries, probability to enter the magazine and latency to enter the magazine). Analyses were conducted separately for each experimental group (ST-Gq, GT-Gi). Bolded values indicate statistical significance, p<0.05.
DOI: https://doi.org/10.7554/eLife.49041.042

• Transparent reporting form
DOI: https://doi.org/10.7554/eLife.49041.043

## Data availability

All data analyzed during this study are included in the manuscript. We also provided source data files for: Figures 2, 3, 4 and 5; for Figure 4—figure upplement1; and for Supplementary files 1–8.

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
