## [Decision Letter]

Thank you for submitting your article "The paraventricular thalamus is a critical mediator of top-down control of cue motivated behavior" for consideration by *eLife*. Your article has been reviewed by three peer reviewers, and the evaluation has been overseen by Geoffrey Schoenbaum as the Reviewing Editor and Kate Wassum as the Senior Editor. The following individuals involved in review of your submission have agreed to reveal their identity: Dona Calu (Reviewer #2); Etienne Coutureau (Reviewer #3).

The reviewers have discussed the reviews with one another and the Reviewing Editor has drafted this decision to help you prepare a revised submission.

Summary:

The authors use chemogenetics to identify a causal role for direct projections from PL to PVT in mediating the balance between sign-tracking and goal-tracking behavior.

Essential revisions:

The reviewers agreed the work was important and provided a nice validation of prior ideas regarding the importance of this pathway. While there are a number of questions and suggestions, none of the revisions were deemed essential for publication provided the authors can provide some information and make a good faith effort to address all the concerns. Assuming this can be done, we do not anticipate the specific outcome of the questions to derail publication.

*Reviewer #1:*

In this paper, the authors test the hypothesis that projections from PRL-PVT are important for regulating the goal-tracking behavior in their ST/GT model. This idea is based on prior findings that cue presentation results in activation of neurons in this pathway in GT but not ST subjects. Here they use a chemogenetic approach to show that activating the pathway results in reduced ST activity in ST subjects (and no reduction in GT subjects which are at floor) while inhibiting the pathway results in increased ST in GT subjects (and no increase in ST subjects – are they at ceiling?). They also examine effects of manipulations on the ability of the lever to support acquisition of a novel instrumental response, and they use microdialysis to examine dopamine release, both with somewhat murky results that did not clearly support their case. But the basic result was novel and adds interestingly to their model.

My main concern as always with this model is that there is little broadening to other behaviors, and the approaches require discarding significant numbers of rats that do not fall into dichotomous categories of ST or GT. This is true here, with the failure of the conditioned reinforcement to produce anything like the expected pattern of results, and the discarding of about 1/3 of the trained subjects because they show intermediate behavior. Both these things make me question the general validity of this behavioral dichotomy. But I am happy to take it at face value as showing a behavior that seems to reflect an abnormal interest in the lever rather than the ultimate location of food delivery.

*Reviewer #2:*

This manuscript investigates the role of PL-PVT for mediating sign and goal-tracking behavioral differences and NAcS dopamine (DA) levels during a Pavlovian Lever Autoshaping (PLA) task. The authors express excitatory (Gq), inhibitory (Gi) or no DREADD receptors in PL-PVT pathway. They give systemic injections of CNO (3 mg/kg) or vehicle to sign-tracking (ST) and goal-tracking (GT) rats prior to reinforced PLA sessions and conditioned reinforcement tests. In Exp.2 NAcSh DA levels are measured during PLA using microdialysis in ST rats expressing Gq DREADDs and GT rats expressing Gi DREADDS in PL-PVT pathway. When excitatory Gq DREADDs are expressed in PL-PVT pathway, relative to vehicle, CNO reduces lever-directed behaviors, increases food cup directed behaviors, prevents conditioned reinforcement in ST rats, but does not alter ST NAcS DA levels. When inhibitory Gi DREADDs are expressed in PL-PVT pathway, relative to vehicle, CNO increases PavCA scores and NAcS DA levels in GT rats, but does not affect responding in conditioned reinforcement tests. When no DREADD receptors are expressed, the authors report no effects of CNO.

The bidirectional manipulation of PL-PVT pathway in both ST and GT rats is an elegant experimental design that tests the authors clearly stated predictions. This manuscript is well written; the data are carefully analyzed and transparently reported. Some clarification is necessary to mitigate potential issues with unequal group sizes and interpretation of conditioned reinforcement effects.

The authors acknowledgement of unequal group sizes and post-hoc power analyses suggest all of the data presented met minimum power needed to reject null hypotheses. There is a notable exception where 1-β is reported for ST group to be marginally underpowered for both behavioral and DA measurements. Is this the only case in which post-hoc power analyses indicated groups were marginally underpowered? Reporting of post-hoc power analyses for Gq GT (n~6 per treatment group) and Gi ST (n~8-9/ treatment group) are most critical. The revision of bolded subtitles should reflect appropriately powered positive and negative effects. Please also report for each tracking group the n's per treatment condition in the Subsection “Subjects”.

Please operationally define conditioned reinforcement. A significant difference between active and inactive pokes? Surpassing some threshold for incentive index? The authors report only the Gq ST VEH group discriminated active/inactive, while Gq VEH ST and no-DREADD ST groups do not. If conditioned reinforcement effects are not consistently observed in ST controls please revise "…top down cortico-thalamic control attenuates the incentive value of a food cue, selectively in ST". The effects of CNO for GT and non-DREADD groups would be more accurately described as "no effects on responding during conditioned reinforcement tests", instead of referring to "conditioned reinforcing properties" of cues for these groups.

*Reviewer #3:*

In this study, the authors investigate the top-down control of cue-motivated behavior. Using the sign-tracker/goal-tracker animal model, the authors found that chemogenetic 1/ stimulation of the prelimbic (PL) to paraventricular nucleus of the thalamus (PVT) pathway decreases the incentive value of a reward cue in sign tracking animals 2/ inhibition of the PL-PVT pathway increases the incentive value of the cue in goal trackers. The topic investigated by the authors is interesting. I think however that additional clarification is needed, as described below.

1) In Experiment 1, it appears that there was a change in phenotype between initial testing and rescreening in a good number of animals (55 out of 246). Could the authors provide information about the nature of the changes (sign to goal or goal to sign tracking responding?). This is important since some of results might be interpreted in terms of normal phenotypic variation across training.

2) I am a bit confused with the final numbers of animals in Experiment 1: n= 126 in the Materials and methods section and then n= 115 (59 GTs vs 56 STs).

3) Research from the same lab (but also from a number of other labs) has provided evidence that phenotypic variations is a very subtle process. I am therefore a bit concerned that Experiment 1 and 2 did not use the same basic experimental setting (Paired vs single-housed, changes in suppliers).

4) I am also a bit confused about the changes in the criteria used for intermediate rats (PavCA index between -0.2 and +0.2 in Experiment 2; PavCA index between -0.3 and +0.3 in Experiment 1). Could the authors explain?

5) Inspection of Figure 1B and Figure 5A suggests that the authors did dissociate aPVT from pPVT in Experiment 1 but not in Experiment 2. Was it the case? If so, this must be clarified in both the methods and Results sections. This is very important since activation (Experiment 1) or inhibition (Experiment 2) could have affected very different portions of the PL-PVT pathway.

6) I also think the authors should better introduce the anatomical features of PL-PVT pathway and cite the corresponding literature.

7) The authors should provide more details about surgery (How was the viruses injected? What was the anesthesia? What was the procedure of pain management before, during and after surgery?)

8) I am a bit confused with the results shown on Figure 3C. Do they mean that sign tracking behavior decreases in the ST CNO even when the CNO is not injected (during rescreening)?

9) There is no doubt that PavCA is a useful index of performance. In the present case however, I think that the data about lever-directed behaviors should be included in the main paper.

10) It could be interesting if the authors could include a discussion about learning vs performance issue.

---

## [Author Response]

[…] Reviewer #1:In this paper, the authors test the hypothesis that projections from PRL-PVT are important for regulating the goal-tracking behavior in their ST/GT model. This idea is based on prior findings that cue presentation results in activation of neurons in this pathway in GT but not ST subjects. Here they use a chemogenetic approach to show that activating the pathway results in reduced ST activity in ST subjects (and no reduction in GT subjects which are at floor) while inhibiting the pathway results in increased ST in GT subjects (and no increase in ST subjects – are they at ceiling?). They also examine effects of manipulations on the ability of the lever to support acquisition of a novel instrumental response, and they use microdialysis to examine dopamine release, both with somewhat murky results that did not clearly support their case. But the basic result was novel and adds interestingly to their model.

To clarify, our prior findings indicate that neurons in the PrL projecting to the PVT are engaged to the same degree in sign-trackers and goal-trackers in response to a food-associated cue (Haight et al., 2017). These findings led us to hypothesize that PrL-PVT neurons encode the “predictive” value of a reward cue and that incentive learning is a function of the subcortical hyperactivity characteristic of sign-trackers. That is, the inherent subcortical activity in sign-trackers can override “top-down” encoding; but activation of the PrL-PVT pathway inhibits this process.

The lack of effect of inhibiting the PrL-PVT pathway in sign-trackers is likely due to a ceiling effect; meaning that there is not much room for further increases on a behavioral level. It should also be considered, however, that this pathway is inherently ‘inhibited’ in sign-trackers.

My main concern as always with this model is that there is little broadening to other behaviors, and the approaches require discarding significant numbers of rats that do not fall into dichotomous categories of ST or GT. This is true here, with the failure of the conditioned reinforcement to produce anything like the expected pattern of results, and the discarding of about 1/3 of the trained subjects because they show intermediate behavior. Both these things make me question the general validity of this behavioral dichotomy. But I am happy to take it at face value as showing a behavior that seems to reflect an abnormal interest in the lever rather than the ultimate location of food delivery.

The basis of these concerns are not entirely clear. It should be noted that the propensity to sign-track has been associated with a number of other behaviors including, greater impulsivity (Lovic et al., 2011), deficits in sustained attention (Paolone et al., 2013), aberrant fear conditioned responses (Morrow et al., 2011; Morrow et al., 2015), and increased propensity for reinstatement of drug-seeking behavior, or relapse (Saunders and Robinson, 2010, 2011). Furthermore, we have consistently shown that a reward-associated cue acts as a conditioned reinforcer to a greater extent for sign-trackers relative to goal-trackers (Robinson and Flagel, 2009; Hughson et al., 2019). Although we did not directly compare sign- and goal-trackers in the current study, these data (Figure 4, Figure 4—figure supplement 1) are consistent with our prior findings.

The focus of the current studies was on the extremes of the population – sign-tracker and goal-trackers – which, indeed, means that we eliminated the intermediate portion of the population (see below). It should be noted, however, that we recently published a manuscript (Hughson et al., 2019) with a population of ~1600 rats for which we included the intermediate population and examined the relationship between the propensity to sign-track and a number of other “addiction-related” behaviors. We found a significant correlation between sign-tracking behavior (i.e. PavCA Index) and behavior during the conditioned reinforcement test (i.e. Incentive Value Index) in this large population of rats; and in the Hughson et al., 2019 manuscript you will find a discussion pertaining to the relationship between these two presumed measures of the incentive value of a reward cue.Reviewer #2:[…] The bidirectional manipulation of PL-PVT pathway in both ST and GT rats is an elegant experimental design that tests the authors clearly stated predictions. This manuscript is well written; the data are carefully analyzed and transparently reported. Some clarification is necessary to mitigate potential issues with unequal group sizes and interpretation of conditioned reinforcement effects.The authors acknowledgement of unequal group sizes and post-hoc power analyses suggest all of the data presented met minimum power needed to reject null hypotheses. There is a notable exception where 1-β is reported for ST group to be marginally underpowered for both behavioral and DA measurements. Is this the only case in which post-hoc power analyses indicated groups were marginally underpowered? Reporting of post-hoc power analyses for Gq GT (n~6 per treatment group) and Gi ST (n~8-9/ treatment group) are most critical. The revision of bolded subtitles should reflect appropriately powered positive and negative effects.

The statistical Results section for the Gq studies has been revised accordingly. The subsection is now:

“Stimulation of the PrL-PVT pathway attenuates the incentive value of the food cue in STs”

We have added the power analysis for GT-Gq and ST-Gi:

“For GTs (Figure 3D), there was not a significant effect of treatment (F_1,10_=0.169, p=0.690), session (F_1,10_=0.511, p=0.491) nor a significant treatment * session interaction (F_1,10_=0.351, p=0.567). It should be noted that the modest sample size (n=6) may have played a role in the lack of effects in GT-Gq, as suggested by a post-hoc power analysis (1-β=0.27). Nonetheless, taken together, these results suggest that “turning on” the top-down PrL-PVT circuit appears to selectively attenuate the incentive value of the cue in STs.”

For Gi, the subsection is now:

“Inhibition of the PrL-PVT pathway increases the incentive vale of the food cue in GTs.”

“For STs (Figure 3E), there was no significant effect of treatment (F1,12=0.876, p=0.368), session (F1,12=4.359, p=0.059) nor a significant treatment * session interaction (F1,12=1.479, p=0.247, 1-β=0.85), suggesting that “turning off” the PrL-PVT pathway permits the attribution of incentive motivational value to a reward-cue selectively in GTs.”

Please also report for each tracking group the n's per treatment condition in the Subsection “Subjects”.

The n’s pretreatment group are now included as follows on pgs.16-17:

“As we were interested only in assessing STs and GTs, rats with a PavCA index between -0.3 and +0.3 (intermediate rats, n= 182) were excluded from the remainder of Experiment 1. Of the remaining 246 rats, 55 were excluded from the analysis because their phenotype changed between the initial testing and rescreening (27 GTs became STs, 26 GTs and 2 STs became intermediate responders). Sixty-five rats were excluded because of inadequate or off-target DREADD expression in the PrL, aPVT or pPVT, resulting in a final n= 126 rats (GT-Gi= 35 (18 VEH/17 CNO), GT-Gq= 12 (6 VEH/6 CNO), ST-Gi= 17 (10 VEH/7 CNO), ST-Gq= 27 (14 VEH/13 CNO), GT-No DREADD= 17 (8 VEH/9 CNO), ST-No DREADD= 18 (8 VEH/10 CNO)). Data for Experiment 1 were collected across 8 rounds of testing.”

“Subjects with inadequate or off-target DREADD expression in the PrL, aPVT or pPVT, or with incorrect microdialysis probe placement (n= 23) in the nucleus accumbens shell were excluded from the statistical analysis, resulting in a final number of 23 rats (GT-Gi= 10 (5 VEH/5 CNO), ST-Gq= 13 (7 VEH/6 CNO)). Technical difficulties during sample collection and/or chemical analysis led to the elimination of 5 ST-Gq samples. Thus, the final number of ST-Gq rats for the microdialysis data was 8 (4 VEH, 4 CNO). Animals from Experiment 2 were tested across 4 different experimental rounds.”

Please operationally define conditioned reinforcement. A significant difference between active and inactive pokes? Surpassing some threshold for incentive index?

We have revised the Results section on conditioned reinforcement such that it now begins as follows.

“A conditioned reinforcement test (CRT) was conducted to assess the reinforcing properties of the lever-CS (Robinson and Flagel, 2009). During this test, responses into a port designated “active” results in the brief presentation of the lever-CS; whereas those in the “inactive” port have no consequence. If a rat responds more into the active port relative to the inactive port, the lever-CS is considered to have reinforcing properties (Robinson and Flagel, 2009). Moreover, if the rat approaches and interacts with the lever-CS during its brief presentation, it is considered to have incentive properties (Robinson and Flagel, 2009; Hughson et al., 2019). Here we use the incentive value index, a composite metric of all behavioral responses during CRT ((pokes in active port + lever-CS contacts) – (pokes in inactive port)), as a primary measure of the conditioned reinforcing properties of the lever-CS (Figure 4; see also Hughson et al., 2019). Nosepoke responding and lever-CS contacts are included in Figure 4.”

The authors report only the Gq ST VEH group discriminated active/inactive, while Gq VEH ST and no-DREADD ST groups do not. If conditioned reinforcement effects are not consistently observed in ST controls please revise "…top down cortico-thalamic control attenuates the incentive value of a food cue, selectively in ST". The effects of CNO for GT and non-DREADD groups would be more accurately described as "no effects on responding during conditioned reinforcement tests", instead of referring to "conditioned reinforcing properties" of cues for these groups.

We apologize for the confusion and would like to clarify that VEH-treated STs and STs in the No-DREADD control groups all respond more in the active port relative to the inactive port during the conditioned reinforcement test. We now do a more thorough job of explaining these results and incorporate the effect of port and any significant interactions into the Results section, in correspondence with Figure 4—figure supplement 1. In addition, per this request, we have changed the subheading and content of the conditioned reinforcement sections as follows.

Subsection “Stimulation of the PrL-PVT pathway attenuates the conditioned reinforcing properties of a reward-cue in STs”: “Stimulation of the PrL-PVT pathway in STs significantly attenuated the incentive value index during the CRT (t_25_ =-3.574, p=0.002, Cohen’s d=1.48, 1-β=0.92, Figure 4b). The same manipulation had no effect on the incentive value index in GTs (Figure 4c). In agreement, there was a significant effect of treatment (F_1,36_ =19.021, p<0.001), port (F_1,36_ =30.501, p<0.001) and a significant treatment x port interaction (F_1,36_=7.024, p=0.012) for nosepoke responding for STs (Figure 4—figure supplement 1a). Post-hoc analysis revealed that, relative to VEH controls, stimulation of the PrL-PVT in STs decreased the number of nosepokes into the active port (p<0.001, Cohen’s d=1.55). Furthermore, while VEH-treated controls responded more in the active port relative to the inactive port (p<0.001, Cohen’s d=2.421), this discrimination between ports was abolished following CNO treatment (p=0.061). Similarly, stimulation of the PrL-PVT pathway reduced the number of lever contacts in STs (t_18.142_ =-3.615, p<0.05, Cohen’s d=1.47, Figure 4—figure supplement 1g). For GTs, there was not a significant effect of treatment, port, nor a significant treatment x port interaction for nosepoke responding (Figure 4—figure supplement 1b), nor a significant effect of treatment for lever-CS interactions (Figure 4—figure supplement 1h). These data are consistent with those reported above for PavCA behavior, as “turning on” this top-down cortico-thalamic control attenuates the incentive value of a food-cue selectively in STs.”

Subsection “Inhibition of the PrL-PVT pathway does not affect the conditioned reinforcing properties of a reward-cue in either STs or GTs”: “Inhibition of the PrL-PVT pathway did not affect the incentive value index for either phenotype (Figure 4d, 4e). For STs, there was a significant main effect of port (F_1,30_ =30.696, p<0.001), for which both VEH- and CNO-treated rats responded more in the active port (Figure 4—figure supplement 1c). For GTs, there was a significant main effect of treatment (F_1,56_ =9.608, p=0.03) and port (F_1,56_ =23.707, p<0.001), but there was not a significant treatment x port interaction (Figure 4—figure supplement 1d). Thus, CNO-treated rats responded more than VEH controls, but they did so across ports (i.e. in both the active and inactive ports); and all GT rats responded more in the active port relative to the inactive port. Inhibition of the PrL-PVT did not affect lever-CS interaction in either phenotype (Figure 4—figure supplement 1, j) during the CRT test. Potential explanations for the seemingly incongruent effects of PrL-PVT inhibition on Pavlovian conditioned approach behavior and the conditioned reinforcing properties of the reward cue in GTs are discussed below.”

Subsection “CNO administration in the absence of DREADD receptors does not affect the conditioned reinforcing properties of a reward-cue in STs or GTs”: “Administration of CNO in the absence of DREADD did not affect the incentive value index during CRT in either STs (Figure 4f) or GTs (Figure 4G). For nosepoke responding, there was a significant main effect for port, for which both STs (F_1,26_ =15.521, p=0.001) and GTs (F_1,26_ =8.492, p=0.007) responded more in the active port relative to the inactive port. There was not a significant effect of treatment, nor a significant treatment x port interaction for either phenotype, suggesting that CNO administration did not affect the conditioned reinforcing properties of the reward cue (Figure 4—figure supplement 1E-F). In agreement, CNO administration did not affect lever-CS interaction during the CRT (Figure 4—figure supplement 1K-L).”

Reviewer #3:[…] 1) In Experiment 1, it appears that there was a change in phenotype between initial testing and rescreening in a good number of animals (55 out of 246). Could the authors provide information about the nature of the changes (sign to goal or goal to sign tracking responding?). This is important since some of results might be interpreted in terms of normal phenotypic variation across training.

We acknowledge the concern that a spontaneous change in phenotype could affect the interpretation of some of the results, which is exactly why we excluded animals whose behavior and resultant phenotype appeared unstable between initial testing (acquisition, sessions 1-5, Figure 2B) and rescreening (left panel of each graph, Figure 3). This subset of animals represented ~22% of our original population. Of these 55 rats, 53 were originally classified as GTs and then showed a tendency towards sign-tracking, with 27 meeting criteria for STs (i.e. PavCA index >0.3 during sessions 9-10), and 26 becoming intermediate responders (i.e. PavCA index between -0.3 and +0.3 during sessions 9-10). Two STs became intermediate responders. These trends support the notion that sign-tracking behavior is stable, persistent and resistant to change (Hearst E., 1974; Tomie, 1996; Ahrens et al., 2016; Chang and Smith, 2016); and also support our previous findings suggesting that, following prolonged training, goal-trackers eventually begin to show interest in the lever-CS (Haight et al., 2015). We believe, however, that different neurobiological mechanisms are regulating the “change in phenotypes” following prolonged training versus the initial acquisition of the conditioned responses (Clark et al., 2013).

Given that we counterbalanced treatment groups and the fact that we are directly comparing the rescreening phase to the test phase in these experiments, we truly do not believe that a tendency for GTs to become STs with prolonged training is affecting our results. In support, we do not see a significant increase in the PavCA index during the test relative to rescreening for either the GT-Gq study or the No-DREADD GT study (Figure 3D,F,H).

Since we eliminated the animals that ‘switched phenotypes’ from this study, we do not believe this discussion warrants attention in the primary text. We have, however, added the following to the subsection “Subjects”:

“Of the remaining 246 rats, 55 were excluded from the analysis because their phenotype changed between the initial testing and rescreening (27 GTs became STs, 26 GTs and 2 STs became intermediate) …”

2) I am a bit confused with the final numbers of animals in Experiment 1: n= 126 in the Materials and methods section and then n= 115 (59 GTs vs 56 STs).

We apologize for the confusion. This discrepancy in the number of subjects is a function of our data analysis. As reported in the subsection “Statistical analyses”, all linear mixed models (LMM) and analysis of variance (ANOVA) were subjected to a Shapiro-Wilk test in order to test the assumption of normality. In most groups (ST-Gq, ST-Gi, GT-Gi, ST-No DREADD, GT- No DREADD), the assumption of normality was violated. In these cases, we employed the commonly-used boxplot method (Tukey, 1977) to graphically explore potential outliers that contributed to data dispersion. The boxplot method uses the interquartile range (IQR) for identifying influential data points that can lead to flawed conclusions when using parametric tests. Data points that fall beyond ± 1.5 times the IQR (i.e. the “inner fence” of the distribution) are considered outliers. Data points that fall beyond ± 3.0 times the IQR (i.e. the “outer fence” of the distribution) are considered extreme outliers. Based on this method, 11 subjects from Experiment 1 (GT-Gq=2; ST-Gi=3; GT-Gi=3; ST-No DREADD=1; GT-No DREADD=2) fell beyond the “inner fences” of the distribution (below 1.5 or above 1.5 times the IQR). These subjects were consequently considered as outliers and excluded from the statistical analysis in order to meet the assumption of normality. For these analyses, therefore, the number of subjects used (n) was 126-11=115. Please note, however (as stated in the main text of the paper), that the exclusion of these subjects did not affect the statistical outcomes (i.e. the significance) of any of the comparisons made. That is, the results were the same with or without the outliers. It should also be noted that, for the independent samples t-tests (e.g. conditioned reinforcement), outliers were not excluded because the t-test is considered to be more robust in the presence of distributions that diverge from normality (Sullivan and Dagostino, 1992). In these cases, therefore, the number of the subjects used for the analysis remained unchanged (n=126).

The relevant section of the text that addresses this concern follows (Subsection “Statistical Analysies”):

“For all LMM and ANOVA analyses, normality was checked using a Shapiro-Wilk test. If normality was violated, the presence of outliers and influential data points was explored using boxplots (Tukey, 1977). Data points that fell below -1.5 or above +1.5 times the interquartile range (IQR) were considered as outliers and excluded from the statistical analysis in order to meet the assumption of normality. Using these criteria, 11 subjects were excluded from Experiment 1 (GT-Gq=2; ST-Gi=3; GT-Gi=3; ST-No DREADD=1; GT-No DREADD=2), resulting in an n=115 for the PavCA index analyses. Importantly, however, the significant results of the statistical analyses were the same with or without the outliers. Graphical representations include only those data without outliers.”

3) Research from the same lab (but also from a number of other labs) has provided evidence that phenotypic variations is a very subtle process. I am therefore a bit concerned that Experiment 1 and 2 did not use the same basic experimental setting (Paired vs single-housed, changes in suppliers).

We acknowledge the importance of using the same experimental settings when studying phenotypic variation based on behavior. However, the differences in housing (paired-housed rats in Experiment 1 vs. single-housed rats in Experiment 2) and in the strain of rats used (Heterozygous Stock in Experiment 1 vs. Sprague-Dawley in Experiment 2) were necessary.

Rats from Experiment 2 had to be single-housed in order to avoid damage to the implanted cannulas that served as a guide for microdialysis probes. Although we have never directly compared single- vs. pair-housed rats and their phenotypic distribution along the sign-tracker/goal-tracker dimensions, we have not observed noticeable differences in the distribution when rats have to be single-housed following surgeries (e.g. jugular catheterization or cannulation).

For Experiment 2, because we were performing surgeries for DREADD expression and cannulation prior to Pavlovian conditioning, we wanted to optimize our chances of getting a relatively equal distribution of sign-trackers and goal-trackers and, at that point, recognized that our chances of doing so were better with outbred rats from select vendors than they were with the heterogeneous stock rats. Thus, it was more efficient, cost-effective and reduced the number of rats used by relying on commercial vendors for Experiment 2. Furthermore, our intent was never to directly compare Experiments 1 and 2; thus, we do not believe the change in housing conditions or strains used diminishes the findings in any way.

4) I am also a bit confused about the changes in the criteria used for intermediate rats (PavCA index between -0.2 and +0.2 in Experiment 2; PavCA index between -0.3 and +0.3 in Experiment 1). Could the authors explain?

This point is addressed above in response to a comment by reviewer #1. Briefly, however, a different cut-off was used for Experiment 2 because the classification of rats into STs or GTs occurred after just 3 sessions of Pavlovian conditioning (as opposed to 5 sessions for Experiment 1), when the conditioned responses are not completely developed. Indeed, after 3 sessions of PavCA training, most of the rats do not reach the cut-off for the criteria used in Experiment 1 (PavCA index ≤-0.3 for GTs and ≥0.3 for STs). Therefore, we used a different cut-off for Experiment 2 (PavCA index ≤-0.2 for GTs and ≥0.2 for STs). This cut-off was chosen because we have found that rats that have a PavCA index ≤ -0.2 or ≥ + 0.20 after 3 sessions of PavCA training have ~90% chance of becoming GTs or STs, respectively, after 5 sessions of training (Koshy Cherian et al., 2017).

5) Inspection of Figure 1B and Figure 5A suggsets that the authors did dissociate aPVT from pPVT in Experiment 1 but not in Experiment 2. Was it the case? If so, this must be clarified in both the methods and Results sections. This is very important since activation (Experiment 1) or inhibition (Experiment 2) could have affected very different portions of the PL-PVT pathway.

For both Experiments we targeted both the anterior PVT (aPVT) and posterior PVT (pPVT). We acknowledge that this was not clear from Figure 1B and 5A and have since revised Figure 5A to include representation of both the aPVT and pPVT, similar to Figure 1B.

We have also added the following sentences for clarification:

Subsection “Acquisition of Pavlovian conditioned approach”: “Based on the results of Experiment 1, Experiment 2 focused on STs expressing Gq DREADD (ST-Gq) and GTs expressing Gi DREADD (GT-Gi) in neurons of the PrL that project to the PVT (both aPVT and pPVT).”

Subsection “Surgery”: “For both Experiments 1 and 2, a dual-vector approach was used to selectively express DREADD receptors in neurons of the PrL that project to the aPVT and pPVT.”

6) I also think the authors should better introduce the anatomical features of PL-PVT pathway and cite the corresponding literature.

We have edited the Introduction to include a brief description of afferent and efferent connections of the PVT with appropriate citations as follows:

“The PVT receives cortical afferents from the medial PFC, including the infralimbic and prelimbic (PrL) cortices, and subcortical afferents from the hypothalamus, amygdala, and several brainstem regions involved in visceral functions and homeostatic regulation (Hsu and Price, 2007; Li and Kirouac, 2012; Kirouac, 2015). The PVT sends projections to various brain regions that have been associated with reward-learning and motivated behaviors, including the PrL and IL cortices, NAc core (NAcC) and shell (NAcS), lateral bed of the stria terminalis and central amygdala (Hsu and Price, 2007; Li andKirouac, 2008; Hsu and Price, 2009). Recent findings surrounding the functional role of these PVT circuits (Do-Monte et al., 2015; Haight et al., 2017; Giannotti et al., 2018) have led to the recognition of the PVT as the “thalamic gateway” (Millan et al., 2017) for appetitive motivation; acting to integrate cognitive, emotional, motivational and viscerosensitive information, and, in turn, guide behavioral responses (Kirouac, 2015; Millan et al., 2017). Consistent with this view, the PVT has been implicated in the propensity to attribute incentive motivational value to reward-cues (Haight andFlagel, 2014; Haight et al., 2015; Kuhn et al., 2018).”

7) The authors should provide more details about surgery (How was the viruses injected? What was the anesthesia? What was the procedure of pain management before, during and after surgery?)

Additional details have been added to the subsection “Surgery”, as follows:

“For both Experiments 1 and 2, a dual-vector approach was used to selectively express DREADD receptors in neurons of the PrL that project to the aPVT and pPVT. Rats were deeply anesthetized using 5% inhaled isoflurane (Isothesia – Butler-Schein, Columbus, OH) and the anesthetic plane was maintained with 2% inhaled isoflurane for the duration of the surgery. Prior to surgeries, while under anesthesia, rats received an injection of carprofen (5mg/kg, s.c.) for analgesia and were further prepared for surgeries by shaving the scalp and applying betadine (Purdue Products, Stamford, CT) followed by 70% alcohol as an antiseptic. Rats were then placed into a stereotaxic frame (David Kopf instruments, Tujunga, CA or Stoelting, Wood Dale, IL) and a small incision was made on the scalp to expose the skull. Two small holes were drilled above the PrL (bilaterally) and above the anterior PVT (aPVT) and posterior PVT (pPVT) using Bregma coordinates as indicated below.”

8) I am a bit confused with the results shown on Figure 3C. Do they mean that sign tracking behavior decreases in the ST CNO even when the CNO is not injected (during rescreening)?

Unfortunately, as groups were not perfectly counterbalanced (because we didn’t know which rats did or did not have accurate DREADD expression at the time of treatment group assignments), there was a significant difference between those rats assigned to the CNO group and those assigned to the VEH group during the rescreening phase, prior to actually receiving treatment. Nonetheless, the results of the statistical analyses indicate that this difference is further pronounced during the test phase, following CNO treatment. That is, CNO-treated rats show a further decrease in PavCA index during the test phase relative to the rescreening phase; whereas those in the VEH-treated group do not differ between the test phase and rescreening phase.

The following text has been added to clarify (subsection “Stimulation of the PrL-PVT pathway attenuates the incentive value of the food cue in STs**”**):

“Post-hoc analyses revealed a significant difference between VEH- and CNO-treated STs during both rescreening (p=0.005, Cohen’s d=1.27) and test (p<0.001, Cohen’s d=2.74). The significant difference between treatment groups during rescreening, prior to actual treatment, is due to the fact that counterbalancing was disrupted once animals were eliminated because of inaccurate DREADD expression. Importantly, however, only the CNO-treated rats exhibited a change in behavior during the test sessions relative to rescreening (p<0.001, Cohen’s d=1.18).”

9) There is no doubt that PavCA is a useful index of performance. In the present case however, I think that the data about lever-directed behaviors should be included in the main paper.

Per this suggestion and those of reviewer #2 above, we have added some text to clarify when changes in the PavCA index were driven primarily by lever-directed vs. magazine-directed behaviors, as follows:

Subsection “Stimulation of the PrL-PVT pathway attenuates the incentive value of the food cue in STs**”:** “Stimulation of the PrL-PVT pathway in STs significantly decreased the PavCA index (Figure 3c), which, in this case, is reflective of a decrease in lever-directed behavior (Supplementary file 3) and an increase in goal-directed behavior (Supplementary file 4).”

Subsection “Inhibition of the PrL-PVT pathway increases the incentive value of the food cue in 202 GTs**”:** “Inhibition of the PrL-PVT pathway in GTs significantly increased the PavCA index (Figure 3f). This effect appears to be driven primarily by a change in the “response bias” score (F_1,30_=4.136 p=0.051, Cohens d=1.04, 1-β=0.99; data not shown), which is a measure of: [(total lever-CS contacts − total food magazine entries) / (total lever-CS contacts + total food magazine entries)] (Meyer et al., 2012).”

Subsection “Inhibition of the PrL-PVT pathway increases the propensity to attribute incentive value to a food cue and DA levels in the NAcS in GTs**”:** Inhibiting the PrL-PVT pathway early in training in GTs resulted in an enhancement of the incentive motivational value of the reward-cue (effect of treatment (F_1,8_=10.722, p=0.011); treatment x session interaction (F_1,8_=7.163, p=0.028; Figure 5i)). This effect was, due, in part, to a decrease in magazine-directed behaviors (Supplementary file 8).”

In addition, all of the statistics regarding these measures can be found in the tables included in the Supplementary files. We believe that lever-directed behaviors are as important as goal-directed behaviors and that the difference between the two may be the most meaningful when assessing changes in “expression” of the behaviors. Thus, we think showing changes in the PavCA index in the primary figures and including detailed statistics for all other measures is the simplest and most effective way to present our results.

10) It could be interesting if the authors could include a discussion about learning vs performance issue.

We edited the Discussion section and added more emphasis on the interpretation of the results in terms of learning vs. performance, as follows:

“This pattern of results is likely due to distinct neural mechanisms regulating behavior during “learning” vs “performance”. […] Ongoing studies will determine whether chemogenetic or optogenetic manipulations of bottom-up processes can similarly alter individual differences in cue-motivated learning strategies.”